# SWE-Ext: Extending and Scaling Augmented Data for Repository-Level Coding Tasks

## Abstract

Repository-level benchmarks such as SWE-Bench have highlighted the challenges of scaling language models to complex software engineering tasks. However, current training data remains narrow in scope, primarily focusing on monolingual issue resolving and feature implementation. In this work, we introduce **SWE-Ext**, a large-scale effort to extend and scale augmented data for repository-level coding tasks. SWE-Ext broadens existing data along two key dimensions: **multilingual coverage** (spanning 10 languages) and an **auxiliary code completion task**. We uncover distinct transfer mechanisms: data from other programming languages provides transferable signals that generally enhance localization and editing capabilities in single-language (Python) settings, while code completion data strengthens code editing capabilities, particularly for feature implementation tasks requiring substantial new code generation. These extensions yield consistent improvements on Python repository-level benchmarks like SWE-Bench and FEA-Bench. Our method offers a simple yet effective way to leverage more open-source data for advancing repository-level code models.

## 1 Introduction

Repository-level coding tasks have emerged as a critical frontier in code generation, shifting focus from isolated script-level challenges to comprehensive software engineering workflows in full repositories. Early benchmarks emphasized standalone function synthesis, such as HumanEval (Chen et al., 2021) for Python problems and MBPP (Austin et al., 2021) for basic algorithms. However, these evaluations often ignored broader repository contexts, including dependencies, multi-file interactions, and real-world specifications (Xu et al., 2022). Recent advancements have introduced more holistic benchmarks, with SWE-Bench (Jimenez et al., 2024) exemplifying repository-level issue resolution by evaluating models on GitHub pull requests (PRs) tied to verifiable test outcomes. Complementary efforts like FEA-Bench (Li et al., 2025) target feature implementation.

To enhance model capabilities on these demanding tasks, substantial work has explored post-training strategies. Broadly, these approaches fall into two categories: augmented data and verified data. Augmented data methods construct training instances directly from real-world GitHub PRs, leveraging ground-truth information without requiring execution environments (Xie et al., 2025; Wang et al., 2025). For instance, SWE-Fixer (Xie et al., 2025) curates PR-based data for instruction tuning, while MCTS-Refine (Wang et al., 2025) synthesizes reasoning chains via search algorithms. In contrast, verified data approaches build executable environments to collect agent trajectories, filtering for those that pass unit tests or achieve successful outcomes (Jain et al., 2025; Pan et al., 2024; Yang et al., 2025b). Techniques like R2E (Jain et al., 2025) and SWE-Gym (Pan et al., 2024) generate validated trajectories in sandbox for real PRs, and SWE-Smith (Yang et al., 2025b) extends synthetic tasks with verification. While verified data ensures high-quality supervision through execution feedback, it demands significant resources for environment setup and scaling. Augmented data, conversely, offers greater flexibility and avoids these complexities, enabling broader exploration of data domains.

Despite their advantages, existing augmented datasets remain limited in scope, predominantly focusing on Python repositories and monolingual issue resolution or feature implementation (Xie et al., 2025). This narrow focus underutilizes the vast diversity of open-source GitHub data, restricting model generalization across languages and task types. In this work, we introduce **SWE-Ext**, a

scalable pipeline to extend and augment repository-level coding data along two orthogonal dimensions: multilingual coverage and auxiliary task inclusion (e.g., code completion). Starting from high-quality PRs crawled from GitHub Archive [1] across ten programming languages (Python, Go, JavaScript, Ruby, PHP, Java, TypeScript, C#, C++, and C), we construct datasets of four complementary sub-tasks: (1) file localization to identify relevant files, (2) component localization to pinpoint functions or methods, (3) code editing to generate patches for code repositories, and (4) code completion derived from short-description PRs to provide code in-filling on fixed-positions. Our contributions are as follows:

- We develop a scalable data extension pipeline that broadens the utility of GitHub PRs for repository-level tasks, enabling extension across multilingual and completion-based domains while preserving baseline effectiveness.

- We demonstrate consistent performance gains on repository-level benchmarks like SWE-Bench and FEA-Bench, with multilingual and completion extensions yielding up to +1.4% and +2.5% improvements on 32B models, respectively, and up to +5.4% on 7B models.

- We provide preliminary validation of strong cross-task and cross-language transfer effects, showing that multilingual or completion data can boost single-language downstream performance and highlighting the value of diverse augmentation for generalization.

## 2 PROBLEM DEFINITION

**Notation.** Let $\mathbb{R}$ denote a software repository represented as a finite collection of files $\{f_i\}_{i=1}^N$ together with auxiliary structures (ASTs, file-level skeletons, import graph, etc.). Let $q$ denote a problem statement (e.g., a PR description, an issue body, a failing test, or a natural-language specification). Let $\mathbb{P}$ be the universe of addressable program positions (file identifiers $\times$ component spans). A *localization* is a subset $\ell \subseteq \mathbb{L}$. The space of edits (patches) is denoted $\mathbb{P}$; an edit (patch) $\delta \in \mathbb{P}$ is an operator $p : \mathbb{R} \to \mathbb{R}'$ that applies insert/replace/delete/move operations to positions in $\mathbb{P}$. Finally, $L(\mathbb{R}', q)$ is a task loss measuring how well the edited repository $\mathbb{R}'$ satisfies $q$. Note that $P(\cdot)$ denotes a probability distribution.

**Repository-level coding task.** Given $(\mathbb{R}, q)$, the objective of a repository-level coding agent is to produce a patch $\delta \in \mathbb{P}$ that minimizes the task loss:

$$\hat{\delta} \ = \ \arg\min_{\delta \in \mathbb{P}} L\big(\delta(\mathbb{R}), \, q\big), \tag{1}$$

or equivalently, under a probabilistic modeling view,

$$\hat{\delta} \ = \ \arg\max_{\delta \in \mathbb{P}} P\big(\delta \mid \mathbb{R}, q\big). \tag{2}$$

**Decomposition into localization and editing.** Because modern LLMs cannot feasibly consume the entirety of large repositories at once, we decompose the posterior over patches by marginalizing over possible localizations:

$$P(\delta \mid \mathbb{R}, q) \ = \ \sum_{\ell \in \mathbb{L}} P(\ell \mid \mathbb{R}, q) \, P(\delta \mid \mathbb{R}, q, \ell). \tag{3}$$

This decomposition separates (i) a *localization* model that identifies candidate regions $\ell$ of interest, and (ii) an *editing* model that produces a concrete patch conditional on the selected regions. In practice, localization and editing are performed in separate iterative phases: the system first iterates to identify a sufficiently accurate set of relevant files or components, and only then initiates an editing phase where patches can be generated, potentially with feedback.

**Code completion as a special case.** Code completion corresponds to the special case where the localization $\ell^\star$ is given *a priori* (a fixed position or contiguous span) and the patch is constrained to be a continuation or replacement at that position:

$$\hat{\delta} \ = \ \arg\max_{\delta \in \mathbb{P}} P\big(\delta \mid \mathbb{R}, q, \ell^\star\big). \tag{4}$$

---

[1] https://www.gharchive.org

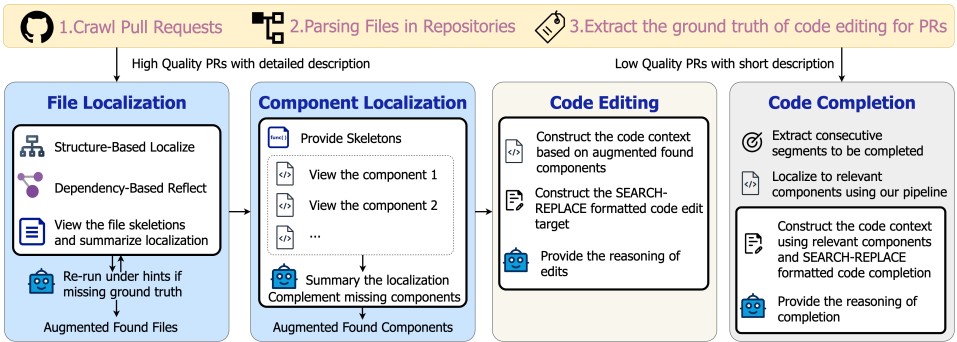

Figure 1: Data collection pipeline for 4 types of tasks in SWE-Ext. We crawl and process pull requests from GH Archive, and then process well described PRs for localization-editing task and process other PRs for completion task.

## 3 SWE-Ext: Extend the Repository-Level Coding Data

### 3.1 Data Collection

**Source data and ground truth.** We crawl pull requests (PRs) from GH Archive (recent 10 years) and retain only high-quality candidate samples. We keep PRs whose repository has both (i) star count $> 100$ and (ii) total PR count $> 100$, and we exclude PRs that modify only non-code files or that were not merged. For each retained PR we collect metadata (PR description, linked issue text if any, timestamps) and patch (diff). We parse repository files with tree-sitter parsers for the following languages: Python, Go, JavaScript, Ruby, PHP, Java, TypeScript, C#, C++, and C. From each file we extract a compact *file skeleton* (top-level classes, functions/methods). Using the patch we record the ground-truth modified files $\mathbb{F}_{gt}$ and the ground-truth modified components $\mathbb{C}_{gt}$. To limit tractability, we retain only PRs with $||\mathbb{F}_{gt}|| \leq 5$ and $||\mathbb{C}_{gt}|| \leq 10$. The problem statement $q$ is the concatenation of the PR description and the linked issue body (when available). Formally each retained example is stored as a tuple

$$s = \left( \mathbb{R}, q, \mathbb{F}_{gt}, \mathbb{C}_{gt}, \delta_{gt} \right), \tag{5}$$

where $\delta_{gt}$ is the ground-truth patch.

**File localization.** To approximate $P(\ell \mid \mathbb{R}, q)$ at file granularity, we construct file-localization data. We employ a three-stage pipeline driven by a expert model: (i) rank files according to the file tree the problem statement; (ii) expand via file-level dependency to form a candidate set; (iii) re-rank using file skeletons and the expert model to produce a final top-$n$ set $\mathbb{S}$. If $\mathbb{F}_{gt} \not\subseteq \mathbb{S}$, we re-run the expert with rationalization hints so that $S$ always covers the ground truth. The resulting dataset pairs $(\mathbb{R}, q, \mathbb{S})$ provide supervision for learning to predict high-recall file sets that contain the true patch locations, thereby reducing the effective search space for downstream editing.

**Component localization.** Once file localization is complete, we build component-level (function/-class level) localization data to approximate $P(c \mid \mathbb{R}, q, \ell)$. We adapt iterative selection methods from CoSIL (Jiang et al., 2025) to allow the expert model to sequentially inspect and select components from candidate files. The process terminates once up to $K = 10$ components are selected. If some ground-truth components are missing, they are inserted into the iterative process and final list. Each example $(\mathbb{R}, q, \mathbb{F}, \mathbb{S}_{comp})$ thus can serves as a high-recall label for a policy that selects components relevant to $q$. This supports models in learning to identify fine-grained patch locations and to maximize recall under constrained inspection budgets.

**Code editing.** We construct code-editing data directly from $\delta_{gt}$. For each selected localization context $\ell_{ctx} = \mathbb{S}_{comp}$, we convert patches into *before/after* code fragments (search–replace pairs $\delta_{sp}$). To enrich supervision, a expert model generates a rationale explaining the patch. Each example $(\mathbb{R}, q, \ell_{ctx}, \delta_{sp})$ provides supervision for $P(\delta \mid \mathbb{R}, q, \ell_{ctx})$. This partition trains models to propose concrete patches given a localized context.

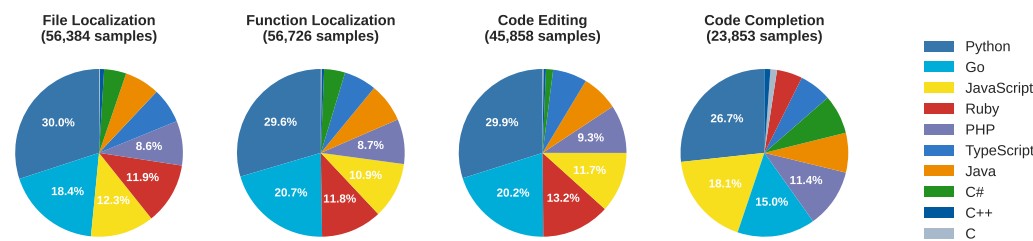

Figure 2: Language distribution of dialogues for 4 sub-tasks in SWE-Ext dataset.

**Code completion.** Finally, we derive a completion dataset from PRs whose problem statements are short (fewer than 50 words), which are possibly not complete instructions for code editing. In these cases, we extract newly added functions or contiguous inserted regions and frame them as completion tasks with a fixed localization $\ell^\star$. Each example provides supervision for $P(\delta \mid \mathbb{R}, q, \ell^\star)$. Because completions originate from real patches and are abundant, they supply local-generation signals that improve fluency and act as an effective curriculum for training editing models.

**Summary.** The construction pipeline yields four complementary datasets, each aligned with a distinct conditional distribution in the factorization of $P(\delta \mid \mathbb{R}, q)$. By supervising different stages separately, we enable scalable training for repository-level coding tasks while ensuring that the final system can integrate these capabilities into a coherent editing pipeline. For further details, please refer to Appendix D.

## 3.2 DATASET CHARACTERISTICS

The SWE-Ext corpus is a *training* dataset constructed from real-world pull requests and intended to supervise models for repository-level coding tasks. The dataset is organized into four complementary tasks: file localization, component (function/method) localization, code editing, and code completion. Table 1 reports core statistics. Below we summarize the most important characteristics and how they relate to model training.

**Origin and construction.** Table 1 presents the statistics of the SWE-Ext dataset. For the three localization and editing tasks (File-Localization, Component-Localization, and Code-Editing), we retain a canonical pool of approximately 56k to 46k PRs. Crucially, these tasks exhibit high data diversity, being sourced from 4,319 to 4,746 unique GitHub repositories, which strongly demonstrates the broad applicability of our data augmentation strategy. The Code-Completion task collects newly-added functions or contiguous insertion ranges from $\sim 23.9$k PRs, whose short descriptions make end-to-end supervision weak but whose insertions are realistic completion targets.

**Augmentation of localizations.** The **Localizations** column in Table 1 highlights our data augmentation strategy: The large candidate set size (e.g., 4.83 files vs. 1.66 ground truth files for File-Localization) confirms that we incorporate highly relevant, non-ground-truth files or components into the training data. This design is crucial because it makes the dataset more realistic, aligning with scenarios where agents' localization is often relevant but not perfectly precise in real-world software engineering tasks.

**Scale, context length and edit complexity.** The dataset provides long, realistic contexts: average input contexts (**Dialogue Len**) range from $\sim 5.9$k tokens (completion) up to $\sim 9.8$k tokens (file-localization), reflecting the long context challenge for real-word repository-level tasks. Average **Answer Len** varies by task ($\approx 60$–955 tokens). Edit complexity differs across modalities: code-editing patches average $\sim 32$ lines (explicit before/after pairs), while completion targets are larger on average ($\sim 88$ lines) because they often correspond to newly-added contiguous code regions.

**Language coverage.** SWE-Ext covers ten languages: Python, Go, JavaScript, Ruby, PHP, Type-Script, Java, C#, C++, and C. Language distributions (See Figure 2) show consistent dominance

Table 1: Statistics of the dialogue data for 4 tasks in the SWE-Ext dataset. Token counts are computed using `Qwen2.5-Coder`. **Dialogue Len** represents the total token count of the full dialogue after applying a chat template. **Answer Len** is the average token count of all "assistant" responses within the dialogue. The 'Answer Len' are the final macro-averaged values. The **Patch** metric shows the average number of lines changed in the ground truth patches. For **Localizations**, the values indicate the number of ground truth modified locations versus the total number of locations in the augmented data.

| Dataset | Unique repos | Samples (#) | Dialogue Len (Avg tok.) | Answer Len (Avg tok.) | Patch (Avg lines) | Localizations (Avg GT/Total) |
|---|---|---|---|---|---|---|
| File-Localization | 4 746 | 56 384 | 9 787.8 | 552.1 | - | 1.66/4.83 |
| Component-Localization | 4 702 | 56 726 | 8 164.3 | 59.8 | - | 2.55/6.32 |
| Code-Editing | 4 319 | 45 858 | 6 447.7 | 955.0 | 32.0 | - |
| Code-Completion | 1 642 | 23 853 | 5 929.7 | 430.1 | 88.0 | - |

of Python and Go across modalities (roughly 27–30% and 15–21% respectively), moderate representation for JavaScript, Ruby and PHP ($\approx$ 9–13%), and smaller but present contributions from Java/TypeScript/C#. Low-resource languages (C, C++) appear at the long tail ($< 1$–1.1% in most partitions). This multi-language composition enables cross-lingual training and evaluation of model robustness while reflecting real-world repository populations.

# 4 EXPERIMENTS

**Training.** To empirically validate whether our augmented data, derived from real-world GitHub PRs, can enhance the proficiency of large language models on complex repository-level coding tasks, and to explore the synergistic effects of incorporating multilingual and code completion data, we employ a supervised finetuning approach tailored for a multi-turn dialogue format.

**Models.** Our methodology leverages GPT-4o (`gpt-4o-2024-05-13`) (Hurst et al., 2024) as a data augmentation expert to transform raw GitHub PR data into a multi-task dataset, as detailed in Section 3.1. For the training phase, we utilize `Qwen2.5-Coder-Instruct` (Hui et al., 2024) as our foundational model. For details of training and models, please refer to Appendix B.

**Agent System.** Our agent system adopts a staged approach. We first perform file and component localization following the same iterative process used for data generation. Subsequent stages, including line-level localization, patch generation, and verification, are executed in an Agentless manner (Xia et al., 2024). We have termed this pipeline **CosAgentless**, signifying the integration of the iterative and fine-grained localization process from CoSIL (Xia et al., 2024) into the standard Agentless inference pipeline.

**Data.** Our dataset is partitioned into a 96% training set and a 4% validation set. For a comprehensive ablation study and a fair comparison with prior works, we define three distinct training configurations. First, the **SWE-Ext-Baseline** model is trained on the standard Python-only data for the first three tasks (localization and editing), consistent with existing agent systems. Second, the **SWE-Ext-Multilingual** model is trained on the first three tasks of our multilingual data. Finally, the **SWE-Ext-Completion** model is trained on Python-only data but across all four tasks. This setup enables a clear analysis of the performance gains derived from extending our data along both multilingual and task-specific dimensions.

**Evaluation.** We evaluate our models on two distinct repository-level coding benchmarks: the well-established SWE-bench (Jimenez et al., 2024) for resolving issues and FEA-Bench (Li et al., 2025) for implementing features. Both the benchmarks only contain Python task instances and the primary metric is the task pass rate (%resolved). For a more granular analysis, we also report the Top-x hit rates, MRR (Mean Reciprocal Rank), and MPP (Mean Precision at Position)(Manning, 2008) for the file and component localization stages. In our evaluation, we use a modified Agentless framework to perform a single complete attempt per task. A task is considered a failure if the process stalls or errors out, resulting in an empty patch.

# 5 RESULTS

## 5.1 SOFTWARE ENGINEERING BENCHMARKS

| Model | System | Expert Model | Exec | % Resolved |
|---|---|---|---|---|
| *Closed Weight Models* | | | | |
| **GPT-4o** (Hurst et al., 2024) | Agentless | - | - | 38.8% |
| **Claude 3.5 Sonnet** (Anthropic, 2024) | Agentless | - | - | 50.8% |
| **Claude 3.7 Sonnet** (Anthropic, 2025a) | SWE-agent | - | - | 58.2% |
| **Claude 4 Sonnet** (Anthropic, 2025b) | SWE-agent | - | - | **72.7%** |
| **Llama3-SWE-RL-70B** (Wei et al., 2025) | Agentless | - | - | 41.0% |
| *Open Weight Models* | | | | |
| **DeepSeek-V3-671B** (Liu et al., 2024) | Agentless | - | - | 42.0% |
| **Kimi K2-1TB** (Team et al., 2025) | Agentless | - | - | **65.8%** |
| **Lingma-SWE-GPT-72B** (Ma et al., 2024) | SWE-SynInfer | - | - | 28.8% |
| **Qwen3-235B-A22B** (Yang et al., 2025a) | OpenHands | - | - | 34.4% |
| **SWE-gym-32B** (Pan et al., 2024) | OpenHands | Hybrid | ✓ | 20.6% |
| **R2E-Gym-32B** (Jain et al., 2025) | OpenHands | Claude 3.5 Sonnet | ✓ | 34.4% |
| **SWE-smith-7B** (Yang et al., 2025b) | SWE-agent | Claude 3.7 Sonnet[*] | ✓ | 15.2% |
| **SWE-smith-32B** (Yang et al., 2025b) | SWE-agent | Claude 3.7 Sonnet[*] | ✓ | 40.2% |
| **SWE-fixer-72B** (Xie et al., 2025) | SWE-Fixer | - | ✗ | 32.8% |
| **SoRFT-Qwen-32B** (Ma et al., 2025) | Agentless | Claude 3.5 Sonnet | ✗ | 30.8% |
| **MCTS-Refine-32B** (Wang et al., 2025) | Agentless | DeepSeek-v3 | ✗ | 32.4% |
| *SWE-Ext Models* | | | | |
| **SWE-Ext-Baseline-32B** | CosAgentless | GPT-4o | ✗ | 31.2% |
| **SWE-Ext-Multilingual-32B** | CosAgentless | GPT-4o | ✗ | 32.6% |
| **SWE-Ext-Completion-32B** | CosAgentless | GPT-4o | ✗ | 32.2% |

Table 2: Resolve rates for existing solutions on **SWE-bench Verified**, collected from (Yang et al., 2025b) and Kimi-K2 (Team et al., 2025) technical reports. **Expert Model** indicates the large language models that generated content during the data construction process. **Exec** indicates whether execution-based feedback is used in the data construction process. All performance numbers are pass@1 (Single attempt using agent systems). [*]Indicates the primary data for training is *mainly* generated by the specified expert model.

| Model | System | % Resolved |
|---|---|---|
| *Zero-Short Inference* | | |
| **GPT-4o** (Hurst et al., 2024) | Agentless | 9.0% |
| **o1** (Jaech et al., 2024) | Agentless | 14.0% |
| **DeepSeek-V3-671B** (Liu et al., 2024) | Agentless-Lite | 11.0% |
| *SWE-Ext Models* | | |
| **SWE-Ext-Baseline-32B** | CosAgentless | 10.0% |
| **SWE-Ext-Multilingual-32B** | CosAgentless | 11.5% |
| **SWE-Ext-Completion-32B** | CosAgentless | 12.5% |

Table 3: Resolve rates for existing solutions on **FEA-Bench Lite**, collected from Li et al. (2025). All performance numbers are pass@1 (Single attempt using agent systems).

**Competitive performance under limited resources.** According to Table 2, although the absolute performance of SWE-Ext models is lower than the strongest closed-weight systems (e.g., Claude 4 Sonnet at 72.7%), our models achieve competitive results among open-weight models even with the baseline data. It is important to note that most higher-performing models either (i) use significantly larger parameter counts, or (ii) are distilled from expert models with higher success rates. In contrast, SWE-Ext only leverages augmented training data without relying on such privileged resources, yet already achieves notable improvements. The results validate our pipeline of data collection.

**Consistent improvements with extended data.** Table 2 and Table 3 demonstrate that the models trained with SWE-Ext consistently outperform the baseline across both benchmarks. On SWE-Bench Verified, our SWE-EXT-MULTILINGUAL/COMPLETION-32B achieve 32.6% and 32.2% resolve rates, respectively, compared to 31.2% of the baseline. On FEA-Bench Lite, similar trends are observed: the model trained with extended data outperform the baseline by up to +2.5%. These results confirm that the proposed data extensions are effective in enhancing model performance.

**Multilingual vs. Completion extension.** We observe distinct patterns between multilingual and completion data extension. Multilingual extension brings more consistent improvements in issue resolving, suggesting that cross-lingual signals help models capture generalizable reasoning strategies. In contrast, completion extension is particularly effective for feature implementation, where generating new functionality requires stronger code completion capabilities. These results highlight that different types of target complementary aspects of repository-level development, and jointly contribute to the overall performance gains.

**Ablation insights.** By comparing the multilingual extension and the completion-based extension, we find that both augmentation strategies yield consistent gains. This suggests that when the amount of original supervision is limited, expanding the dataset in orthogonal directions, either through multilingual variants or through completion-style tasks, can provide complementary signals and enhance model generalization. This result underscores the value of leveraging diverse forms of augmented data for repository-level coding tasks.

Overall, these findings highlight that SWE-Ext contributes a practical approach for improving the software engineering capabilities, providing consistent benefits across different benchmarks.

## 5.2 LOCALIZATION ANALYSIS

| Model | System | Component-level Localization | | | | File-level Localization | | | |
|---|---|---|---|---|---|---|---|---|---|
| | | Hit@1 | Hit@3 | MAP | MRR | Hit@1 | Hit@3 | MAP | MRR |
| **Qwen2.5-Coder-32B** | CoSIL | 43.0 | 54.3 | 46.1 | 48.9 | 60.7 | 77.3 | 69.8 | 69.4 |
| **Qwen2.5-Coder-32B** | CosAgentless | 47.4 | 61.8 | 47.4 | 54.8 | 69.0 | 86.2 | 76.1 | 77.7 |
| **SWE-Ext-Baseline** | CosAgentless | 55.0 | 65.4 | 53.0 | 60.8 | 72.8 | 87.6 | 77.0 | 80.2 |
| **SWE-Ext-Multilingual** | CosAgentless | **57.2** | **68.2** | **55.3** | **63.1** | **75.8** | **90.2** | **79.4** | **82.8** |
| **SWE-Ext-Completion** | CosAgentless | 52.8 | 62.0 | 51.3 | 57.9 | 72.8 | 86.6 | 76.7 | 79.8 |

Table 4: Localization performance on the **SWE-bench Verified** test set, consolidating Component-level and File-level results. All scores are reported in percentages (%).

To better understand how augmented data improves final resolution rates, we analyze the intermediate step of *localization* on SWE-bench Verified. Localization accuracy determines whether the correct files or components are identified, and thus directly affects downstream success.

**Effect of multilingual extension.** Introducing multilingual data consistently improves localization: component Hit@1 increases from 55.0% to 57.2%, and file Hit@1 from 72.8% to 75.8% (Table 4). This shows that data from other programming languages provides transferable structural and semantic signals that strengthen the model's ability to locate relevant code regions. As a result, localization accuracy and overall resolution both improve, even when the target task is restricted to Python repositories.

**Effect of completion extension.** In contrast, extending with completion data leads to a different effect: component Hit@1 decreases from 55.0% to 52.8%, while file-level metrics remain nearly unchanged. This is expected, since completion data focuses on the code editing objective rather than localization. Formally, such data trains the model according to Eq. 4, which optimizes the synthesis of code edits $\delta$ given a fixed ground-truth location $\ell^\star$, without providing supervision for selecting $\ell$. Despite the degradation in localization, the strengthened editing capability yields higher overall resolution rates, demonstrating that the positive impact on code edit outweighs the negative impact on localization.

**Implication.** These results reveal distinct mechanisms: multilingual extension simultaneously improves localization and editing by enabling transfer across languages, whereas completion extension primarily enhances code editing ability, especially in scenarios that require substantial additions of new content, such as feature implementation.

### 5.3 SCALING ANALYSIS

To validate the effectiveness and robustness of our data construction approach, we conduct scaling experiments across both model sizes and training data volumes. We evaluate our method on `Qwen2.5-Coder-7B-Instruct` and systematically sample 20%, 40%, and 100% of our augmented training data to examine scaling behaviors. The results are presented in Figure 3.

**Consistent scaling patterns across model sizes.** Our experiments reveal that scaling laws hold consistently across different model capacities. The 32B model substantially outperforms its 7B counterpart across all configurations, achieving an average improvement of +12.8%. Notably, the relative performance trends between different extension strategies remain remarkably stable: multilingual extension consistently provides the largest gains (+5.4% for 7B, +1.4% for 32B over baseline), while completion extension shows modest improvements. This consistency across model scales validates the robustness of our data construction pipeline and suggests that both multilingual and completion extensions scale predictably with model parameters.

**Extension strategies enhance data scaling.** Figure 3(a) shows that applying our extension strategies yields clear logarithmic scaling patterns as the training set size increases. The largest gains (+9.4%) appear when moving from 0% to 20% of the data, with diminishing returns thereafter. Compared with the SWE-Ext-Baseline, which simply discards non-Python samples or PRs with omitted description and achieves only 18.4% on SWE-bench Verified, our approach lifts the entire scaling curve. By converting previously unusable data into effective training signals through multilingual and completion extensions, we substantially improve both data efficiency and the scaling potential of repository-level code generation models.

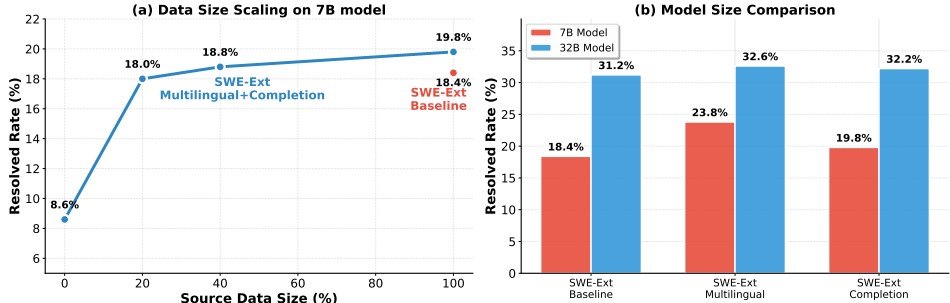

Figure 3: Scaling law analysis on SWE-bench. (a) Performance scaling with training data size follows an approximate logarithmic curve. (b) Model size comparison across three configurations demonstrates consistent improvements from 7B to 32B models.

## 6 RELATED WORK

### 6.1 REPOSITORY-LEVEL CODING TASK

Early benchmarks for code generation focused on function-level tasks, such as HumanEval (Chen et al., 2021) and MBPP (Austin et al., 2021) for simple algorithms. These evaluations emphasized isolated synthesis but overlooked repository contexts. Subsequent benchmarks introduced class- and multi-file challenges, including ClassEval (Du et al., 2023), BigCodeBench (Zhuo et al., 2024), and LiveCodeBench (Jain et al., 2024). Evaluation of real problem in cpde repositories is advanced with SWE-bench (Jimenez et al., 2024) for GitHub issue resolution, alongside variants like DevEval (Li et al., 2024b) and EvoCodeBench (Li et al., 2024a) that align with real repositories. FEA-Bench (Li et al., 2025) targets feature additions via pull requests. Our work aims to enhance the capabilities

of large language models in repository-level coding tasks through post-training with augmented real-world data.

## 6.2 SOFTWARE ENGINEERING AGENTS

Although basic code large language models (Hui et al., 2024) can achieve good performance on many code benchmarks through direct generation, for repository-level tasks involving numerous files and complex edits, the common practice is to incorporate agent frameworks (Yao et al., 2023). Agents for repository-level tasks often employ iterative processes for issue resolution. SWE-agent (Yang et al., 2024) uses agent-computer interfaces for navigation and editing, while AutoCodeRover (Zhang et al., 2024) integrates fault localization for repairs. CodePlan (Bairi et al., 2024) focuses on planning modifications. Localization-specific agents include CoSIL (Jiang et al., 2025), and LocAgent (Chen et al., 2025) for multi-hop reasoning via heterogeneous graphs. Agentless (Xia et al., 2024) simplifies to a three-phase process without complex tooling. Multi-agent platforms like OpenHands (Wang et al., 2024) facilitate collaboration. Our data construction method draws from the commonalities of these agents, employing a pipeline to build data for four subtasks, thereby enhancing model capabilities in repository-level coding.

## 6.3 TRAINING DATA FOR REPOSITORY-LEVEL CODING

Traditional instruction tuning methods enhance code models' adaptability for general code generation, such as WizardCoder (Luo et al., 2023), WaveCoder (Yu et al., 2024), and Magicoder (Wei et al., 2024). However, constructing training data for repository-level tasks differs significantly, as it requires selectively building input-output and reasoning processes to handle complex interactions.

Mainstream approaches involve building executable environments to collect agent trajectories for rejection sampling and supervised fine-tuning. R2E (Jain et al., 2025) and SWE-Gym (Pan et al., 2024) create runtime environments to gather verified trajectories from limited task instances. SWE-Smith (Yang et al., 2025b) extends synthetic data generation to produce more verifiable tasks under constrained environments. Other methods focus on reinforcement learning-compatible data. SWE-RL (Wei et al., 2025) refines reasoning using software evolution data, while SoRFT (Ma et al., 2025) employs subtask-oriented fine-tuning with rejection sampling and PPO.

Due to the diversity requirements of code repositories, some works augment real-world data or synthetic data without constructing environments. SWE-Fixer (Xie et al., 2025) gathers data with chain-of-thought, and MCTS-Refine (Wang et al., 2025) builds reasoning chains via MCTS to form instruction data. In relation to these efforts, while training data construction for repository-level tasks predominantly focuses on Python, our SWE-Ext pipeline extends the data scope by systematically gathering multi-language data of different subtasks, demonstrating that even out-of-distribution data like code completion can further enhance model performance in repository-level scenarios.

## 7 CONCLUSION

In this work, we introduced SWE-Ext, a scalable pipeline for extending augmented data in repository-level coding tasks. By broadening coverage to multilingual PRs across ten languages and incorporating code completion as an auxiliary task, we expand the scope of GitHub-derived training data beyond traditional Python-centric datasets. Our experiments demonstrate consistent improvements on benchmarks like SWE-bench and FEA-Bench, with multilingual extensions enhancing overall abilities and completion data strengthening code editing. Moreover, we reveal robust cross-task and cross-language transfer, where diverse data sources benefit monolingual performance, underscoring the potential of data extension in different dimensions for model generalization.

**Limitations and future directions.** SWE-Ext primarily focuses on augmented data without integrating execution-based verification, which could further refine supervision quality. Additionally, due to the constraints on computational and API resources, we cannot use better expert models for data construction and carry out more experiments for more data combinations and on more base models. Future work could explore verifiable pipelines or extend to additional languages and task types, such as refactoring or generation of unit tests.

ETHICS STATEMENT

We are committed to the responsible and ethical development of artificial intelligence. Our source data is derived exclusively from publicly available, open-source code on GitHub, which inherently poses minimal privacy risks.

During the data augmentation process, we acknowledge the potential for the expert model to generate content that could be considered harmful. However, as these outputs are constrained to code-related reasoning and dialogues, the risk is considered manageable.

As is the case with any model trained on software engineering data, the models trained on our dataset may produce outputs containing security vulnerabilities or even malicious code. Executing these outputs carries a risk of compromising a system. Therefore, for all evaluation and deployment purposes, we strongly recommend that users operate within a securely isolated environment, such as a Docker container, to mitigate any potential harm.

REPRODUCIBILITY STATEMENT

We are committed to ensuring the reproducibility of our results. To facilitate the replication of our work, we provide a comprehensive description of our methodology and experimental setup. In Appendix D, we detail the prompts and the full pipeline used for data construction. Furthermore, Appendix B provides all specified hyperparameters and configurations of our training, which were conducted using a standard open-source training framework. For further reference and insight into the data format, representative data examples are included in the supplementary materials. This detailed documentation provides all the necessary components for researchers to reproduce our findings.

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

## A    USE OF LLMs

Large Language Models (LLMs) were applied in this research in two specific ways. First, LLMs assisted in the development of the codebase, primarily through code completion functionalities. Second, LLMs were employed for language refinement to improve the clarity and readability of the manuscript. All research ideas, experimental designs, and methodological frameworks were independently conceived and implemented by the authors. The authors take full responsibility for the content and conclusions presented in this work.

## B    TRAINING SETTINGS

From our preliminary experiments, we found that using `Qwen2.5-Coder` as the base model yields strong performance, consistent with prior works on constructing training datasets (Yang et al., 2025b; Jain et al., 2025; Wang et al., 2025). We adopt `Qwen2.5-Coder-32B-Instruct` and `Qwen2.5-Coder-7B-Instruct` as the backbone models for fine-tuning. For the training framework, we use the multi-turn dialogue SFT framework provided by `verl` (Sheng et al., 2024), which is implemented on top of FSDP (Zhao et al., 2023).

For the 32B model, we fine-tune on two GPU workstations, each equipped with 8 NVIDIA H200 GPUs. Training is performed in `float32` precision with CPU offloading enabled. We set the maximum sequence length to 32,768. The training configuration includes a sequence parallel size of 2, a learning rate of $5 \times 10^{-6}$, a training batch size of 128, and training for 3 epochs. The total training time is approximately 2, 7, and 3 days on the SWE-EXT-BASELINE, SWE-EXT-MULTILINGUAL, and SWE-EXT-COMPLETION datasets, respectively.

For the 7B model, we fine-tune on a single workstation with 8 NVIDIA A100 GPUs. The setup remains the same as the 32B model, except that the sequence parallel size is increased to 4. The training time is similar to that of the 32B model.

## C    EVALUATION SETTINGS

All execution-based evaluations were conducted on an cloud computing instance equipped with a 32-core AMD EPYC 7763 processor @ 2.45GHz and 256 GB RAM, running Ubuntu 22.04.5 LTS with Linux kernel 6.8.0-1027-azure. It is important to note that in our evaluation environment, we encountered compatibility issues with certain test cases where 5 out of 500 gold patches in SWE-bench Verified and 9 out of 200 gold patches in FEA-Bench Lite failed to execute successfully due to environment-specific dependencies or configuration conflicts. As a result, the reported performance metrics may represent a slight underestimate of the true capabilities of our approach.

## D    PROMPTS OF COSAGENTLESS AND DETAILS OF AUGMENTED DATA

In this section, we detail the prompts used and the specifics of how they construct our four task datasets. All of our data is built upon real-world GitHub pull requests and augmented based on

their extracted ground truth results, which is why we refer to it as "**augmented data**". This data augmentation process consists of transforming the raw pull requests into a dialogue-based format, following the steps of our CosAgentless pipeline.

## D.1 FILE LOCALIZATION

**Phase 1: File Localization.** The first phase of the task is to select a top-5 list of relevant files based on the problem statement (`problem_statement`) and the provided file tree (`structure`). The prompt used for this task is as follows:

```
The problem description is as follows:
```
### GitHub Problem Description ###
{problem_statement}

###

### Candidate Files ###
{structure}

###
```
Let's locate the relevant file step by step using reasoning.
In order to locate accurately, you can pre-select {pre_select_num} files,
 and finally confirm {top_n} file names.

Based on the available information, confirm and provide complete name of
the top-5 most likely relevant files that need to be edited for the
problem.
You should output your reasoning process first.
Since your final answer will be processed automatically, please give your
 final answer of relevant files in the format as follows.
The returned files should be separated by new lines ordered by most to
least important and wrapped with ```.
```
file1.py
file2.py
file3.py
file4.py
file5.py
```
Replace the 'file1.py' with the actual file path.
For example,
```
sklearn/linear_model/__init__.py
sklearn/base.py
```
```

**Phase 2: Dependency Analysis and Reflection.** The second phase analyzes the dependencies of the files selected in the previous stage based on their import relationships `import_content`. These relationships are derived from code segments identified by regular expressions. The model is then prompted to reflect its selection and choose a new list of up to 10 relevant files.

```
Please look through the following problem description and repository
structure and provide a list of files that one would need to edit to
solve the software development problem.
I have already find 5 relevant files. Accrording to the import relations,
 construct the call graph first.

### Problem Description ###
{problem_statement}
```

```
###

### Repository Structure ###
{structure}

###

### Files To Be Explored ###
{pre_files}

###

### Import Relations ###
{import_content}
###

Based on the import relationships, please analyze which files in the
repository depend on which other files within the same repository. Ignore
 any libraries or modules that are imported from outside the current
repository. Present the results in the following format:

file1.py -> file6.py, file7.py
file2.py ->
file3.py -> xxx/file8.py
file4.py -> file2.py, file9.py
file5.py -> xxx/file10.py, xxx/file11.py

Note: Solving the problem not only requires determining where to modify
the code, but also identifying which other code to refer to in order to
understand and invoke.

Based on the files listed above and the import relations, reconfirm and
provide the complete names of the top 10 most likely relevant files,
considering both where changes need to be made and which files are
important to refer to.

Please think step by step and give you reasoning process first. Finally,
provide full path and return top 10 files.

The final returned files should be separated by new lines ordered by most
 to least important and wrapped with ```
For example:
```
file1.py
file2.py
file3.py
file4.py
file5.py
file6.py
file7.py
file8.py
file9.py
file10.py
```
Note: file1.py indicates the top-1 file, file2.py indicates the top-2
file, and so on. Do not include test files.
```

**Phase 3: Refined File Localization.** This phase leverages the results of parsing code repository. The model is provided with the structural skeletons (file_internal_structure, including class and function definitions) of the files identified in the previous stage. Using the skeletons of the files, the model makes a final selection, outputting a refined list of at most 5 files as the final localization result.

```
Please look through the following problem description and repository
structure and provide a list of files that one would need to edit to
solve the software development problem.
I have already find 10 relevent files according to the file structure and
 dependency.
I will further give the internal structures of these 10 files.
Please rank them again and reflect the result according to the internal
structures and dependency.

### Problem Description ###
{problem_statement}

###

### Import Relations ###
{import_content}

###

### Files To Be Ranked ###
{file_internal_structure}

###

Please think step by step and give you reasoning process first. Finally,
provide full path and return top 5 files.

The returned files should be separated by new lines ordered by most to
least important and wrapped with ```
For example:
```
file1.py
file2.py
file3.py
file4.py
file5.py
```
Note: file1.py indicates the top-1 file, file2.py indicates the top-2
file, and so on. Do not include test files.
```

The CosAgentless inference pipeline is structured as a sequential execution of the three phases detailed above. However, during the construction of our ground-truth-based augmented dataset, we employ a specific self-correction process: if the expert model fails to recall all the files in the ground truth code edit in its initial attempt, we provide the ground truth files as an explicit prompt to guide a re-prediction.

## D.2 COMPONENT LOCALIZATION

The prompt for our component localization process is an enhanced version of the one presented in Jiang et al. (2025), specifically adapted to generalize to a broader set of programming languages and repository-level coding tasks.

**System Prompt.** Within the system prompt, we define a set of tool-use capabilities that enable the model to inspect the code of a specified component.

```
You will be presented with a repository-level coding problem with
repository file structure to access the source code of the software.
Since the modification is based on the code repository, the modified
locations may include files, classes, and functions, and the
modifications may be in the form of addition, deletion, or update.
```

```
Your task is to locate the top-5 most likely edit locations based on the
problem description and the information you retrieve using given
functions.
Function calls you can use are as follows:
* get_code_of_class('file_name', 'class_name') -> Get the code of a
specified class in the given file and python project. 'file_name' -> The
name of the file. 'class_name' -> The name of the class. *
* get_code_of_class_function('file_name', 'class_name', 'func_name') ->
Get the code of a specified function in the given class, file, and python
 project. 'file_name' -> The name of the file. 'class_name' -> The name
of the class. 'func_name' -> The name of the function. *
* get_code_of_file_function('file_name', 'func_name') -> Get the code of
a specified function in the given file and python project. 'file_name' ->
 The name of the file. 'func_name' -> The name of the function. *
* get_toplevel_code('file_name') -> Get all the code in a given file that
 is not part of a class or function definition. This is useful for
viewing imports, global variables, constants, and any top-level script
logic. 'file_name' -> The name of the file. *
* exit() -> Exit function calling to give your final answer when you are
confident of the answer. *
You have {max_try} chances to call function.
```

**Iterative Localization Initialization.** The iterative localization process begins with an initialization step where the model is prompted to identify a single, relevant component. This component serves as the initial point of inspection for the codebase.

```
### Problem Description ###
{problem_statement}

###
Let's locate the relevant elements (function/class) step by step using
reasoning and function calls.
I have pre-identified top-5 relevant files. There stuctures are as
follows:
{bug_file_list}
The formal parameter 'file_name' takes the value in "file:"
The formal parameter 'class_name' takes the value in "class:"
The formal parameter 'func_name' takes the value in "static functions:"
and "class functions: "
Avoid making multiple identical calls to save overhead.
You must strictly follow the structure I give to call different tools.
For static functions, you can use 'get_code_of_file_function', and for
class functions, you can use 'get_code_of_class_function'.
In order to locate accurately, you can pre-select {pre_select_num}
locations, then check them through function calls, and finally confirm {
top_n} file names.
Don't make the first function call in this message.
```

**Iterative Component Inspection.** This process involves the iterative inspection of components. At each step, the model leverages the full dialogue history to inform its decision. By making sequential function calls, it dynamically identifies the next component or code segment to examine, progressively refining its understanding of the problem and the codebase. file_internal_structure are the skeletons of found files in the file localization stage.

```
Now call a function in this format 'FunctionName(Argument)' in a single
line without any other word or signal (such as ''').
Don't call the same function you've previous called, because this may
waste your context length.
{file_internal_structure}
```

For each component `component_retrieved` viewed during an iteration, the model is prompted to assess its relevance to the given problem. If the component is relevant to the problem description, the model is then directed to consider its internal call and dependency relationships to select the next component to be viewed. Conversely, if a component is determined to be irrelevant, no further analysis is conducted on it.

```
You will be presented with a repository-level coding problem with
repository file structure to access the source code of the software.

Your task is to locate the top-5 most likely edit locations based on the
problem description.

### Problem Description ###
{problem_statement}

###

Here is a result of a function/class code retrived by '{content}'.
Please check if the code is related to the problem and if the code should
 be added into context.

{component_retrieved}

Return True if the code is related to the problem and should be added
into context, otherwise return False.
Since your answer will be processed automatically, please give your
answer in the format as follows.
The returned content should be wrapped with '''.
'''
True
'''
or
'''
False
'''
```

**Final Component Localization Output.** The final component localization output is generated upon the termination of the iterative process. This occurs when the model either invokes the `exit()` function or reaches the maximum iteration limit. The model is then instructed to summarize its full inspection history and produce a final set of predicted components relevant to the `problem_statement`.

```
{file_internal_structure}
Based on the available information, reconfirm and provide complete names
of the most likely edit locations (10 locations at most).
Before make the final decision, please check whether the function name is
 correct or not, for static functions, don't add class name.
{bug_file_list}

Please provide the complete set of locations as either a class name, a
function name, or a file name.
The returned files should be separated by new lines ordered by most to
least important and wrapped with '''
Since your answer will be processed automatically, please give your
answer in the exapmle format as follows.
'''
top1_file_fullpath.py
function: Class1.Function1

top2_file_fullpath.py
function: Function2

top3_file_fullpath.py
```

```
class: Class3

top4_file_fullpath.py
function: Class4.Function4

top5_file_fullpath.py
function: Function5

top6_file_fullpath.py
global

top7_file_fullpath.py
function: Class7.Function7
```
Replace the 'Top_file_fullpath.py' with the actual file path, the 'Class'
 with the actual class name and the 'Function' with the actual function
name. 'global' means the code is not in a class or function.
For example,
```
sklearn/linear_model/__init__.py
function: LinearRegression.fit
```

During the construction of our augmented data from GitHub pull requests, we incorporate a feedback mechanism. If the final localization output does not fully recall all of the locations specified by the ground truth code edit, we explicitly introduce the omitted locations back into the iterative inspection process. This allows the model to correct its course and regenerate a complete and accurate final localization result.

### D.3 CODE EDITING

**Final Code Editing.** Leveraging the code at the locations identified by our component localization process, we construct a context block denoted as `top_n_content`. This contextual information is then provided to the large language model to facilitate the code editing task.

```
We are currently solving the following task within our repository. Here
is the task description.

### Task Description ###
{problem_statement}

###

Below are some code segments, each from a relevant file. One or more of
these files may need to be edited to solve the task.

--- BEGIN FILE ---
```
{top_n_content}
```
--- END FILE ---

Please first localize the positions to edit based on the task statement,
and then output the files that need to be deleted, modified or added.
'- file' means deleting the file;
'* file' means modificating the file;
'+ file' means adding the file.
The file should list like below:
```
- file1.py
* file2.py
```

```
* file3.py
+ file4.py
```

To solve the task, you should then generate *SEARCH/REPLACE* edits.

Every *SEARCH/REPLACE* edit must use this format:
1. The file path
2. The start of search block: <<<<<<< SEARCH
3. A contiguous chunk of lines to search for in the existing source code
4. The dividing line: =======
5. The lines to replace into the source code
6. The end of the replace block: >>>>>>> REPLACE

Here is an example:

```python
### mathweb/flask/app.py
<<<<<<< SEARCH
from flask import Flask
=======
import math
from flask import Flask
>>>>>>> REPLACE
```

Please note that the *SEARCH/REPLACE* edit REQUIRES PROPER INDENTATION.
If you would like to add the line '        print(x)', you must fully
write that out, with all those spaces before the code!
Wrap the *SEARCH/REPLACE* edit in blocks ```python...```.
When multiple edits should be done, please output *SEARCH/REPLACE* edit
one by one and give your reasoning process before each *SEARCH/REPLACE*
block.
```

**Code Editing Output Format.**    The model's output is structured according to a specific format. The code edits are presented as **SEARCH-REPLACE** pairs, where the old code is provided alongside the corresponding new code. In addition, the model is required to generate a detailed editing plan and a clear rationale for the proposed changes.

```
The plan of solving this software task:
{plan}
The files that should be edited include:
{files}

Here are my edits for code.
{search_replaces}
```

For the construction of our code editing dataset, we first transform the ground truth code edits into a structured format of **SEARCH-REPLACE** pairs. Subsequently, we utilize the expert model to generat the reasoning for these edits, populating the designated `plan` field of the response.

## D.4    CODE COMPLETION

The code completion data is constructed from pull requests that lack a complete problem description ($< 50$ words). When generating these samples, we prioritize targeting a complete, newly added function within the patch. If no such function exists, the target becomes a contiguous block of new code. The code in the context (`top_n_content`) is then composed of the target file along with other relevant components identified via our CosAgentless localization pipeline. The line requiring completion is replaced with the token `[TODO]`, thereby converting the code completion task as an editing task centered on the `[TODO]` token. Consequently, the prompt and the subsequent augmented data generation steps align with those used for our code editing task.

# E DETAILS OF FILTERING

## E.1 FILTERING PULL REQUESTS

To ensure the collection of high-quality data during the pull request (PR) scraping phase, we applied the following exclusion criteria:

- PRs originating from repositories with fewer than 100 stars or a total of fewer than 100 PRs, as this often indicates low repository quality or inconsistent contribution patterns.
- PRs that were not merged into the main branch, indicating that the proposed code changes were not accepted.
- PRs whose patches failed to apply cleanly to the codebase.
- PRs where `tree-sitter` encountered parsing errors in the changed files.
- PRs whose changed files exclusively consisted of non-code file extensions, including: `[".json", ".png", "csv", ".txt", ".md", ".jpg", ".jpeg", ".pkl", ".yml", ".yaml", ".toml"]`.

## E.2 FILTERING TRAINING DATA

After the construction of the SWE-Ext training data, further filtering is required to ensure data quality. First, we remove data samples that contain expert model API call errors, which are commonly caused by exceeding the context window length. Second, we remove samples with obvious errors for each data type:

- **File Localization data**: We remove samples where the ground truth edited files are not fully contained within the final prediction or are not included in the Phase 2 results.
- **Component Localization data**: We remove samples where the ground truth edited components are not fully contained in the final output.
- **Code Edit and Code Completion data**: We remove samples where the "search" code (the code to be changed) is not present in the provided context, as well as those with incomplete search-replace pairs.

To ensure the model is trained on complete conversational data, we also filter out any samples with a length exceeding 32768 tokens. Finally, we remove any dialogue data that corresponds to pull requests included in our evaluation datasets, including SWE-bench Verified and FEA-Bench, to prevent data leakage.

# F DATA EXAMPLES

In this section, we present several short data examples for reference. Each turn of the dialogue is formatted as follows:

```
>>>>>>>>>>>>>>>>>>>>>{role}
{content}
```

## F.1 FILE LOCALIZATION

```
>>>>>>>>>>>>>>>>>>>> system:

You will be presented with a repository-level coding problem with
repository file structure to access the source code of the software.
Your task is to locate the most likely relavant files that need to be
edited based on the problem description.

>>>>>>>>>>>>>>>>>>>> user:
```

```
The problem description is as follows:
```
### GitHub Problem Description ###
<request>
Adding fill parameters to pairwise and windowed
I found these handy for when I needed lookahead/lookbehind, but didn't
want to add extra processing code before/after the processing loop. Here'
s a sketch of one way I've used it:

```python
for record, next_record in pairwise_iter(records, fill=None):
  if next_record is not None:
      record.link(next_record)
  record.process()
```

Without fill, I have to call `records[-1].process()` after the loop,
assuming it's a list and not an iterable.
</request>

###

### Candidate Files ###
boltons/
    __init__.py
    cacheutils.py
    debugutils.py
    deprutils.py
    dictutils.py
    easterutils.py
    ecoutils.py
    excutils.py
    fileutils.py
    formatutils.py
    funcutils.py
    gcutils.py
    ioutils.py
    iterutils.py
    jsonutils.py
    listutils.py
    mathutils.py
    mboxutils.py
    namedutils.py
    pathutils.py
    queueutils.py
    setutils.py
    socketutils.py
    statsutils.py
    strutils.py
    tableutils.py
    tbutils.py
    timeutils.py
    typeutils.py
    urlutils.py
docs/
    conf.py
misc/
    bench_omd.py
    linkify_changelog.py
    table_html_app.py
setup.py

###
```
```

```
Let's locate the relevant file step by step using reasoning.
In order to locate accurately, you can pre-select 7 files, and finally
confirm 5 file names.

Based on the available information, confirm and provide complete name of
the top-5 most likely relevant files that need to be edited for the
problem.
You should output your reasoning process first.
Since your final answer will be processed automatically, please give your
 final answer of relevant files in the format as follows.
The returned files should be separated by new lines ordered by most to
least important and wrapped with ```.
```
file1.py
file2.py
file3.py
file4.py
file5.py
```
Replace the 'file1.py' with the actual file path.
For example,
```
sklearn/linear_model/__init__.py
sklearn/base.py
```

>>>>>>>>>>>>>>>>>>>> assistant:
To solve the problem of adding fill parameters to `pairwise` and `
windowed` functions, we need to identify where these functions are likely
 implemented. The problem description suggests that these functions are
related to iterating over collections, which is a common task in utility
modules dealing with iteration or functional programming.

Given the directory structure, the most relevant file is likely to be `
iterutils.py` because:
- The name `iterutils.py` suggests it contains utilities for iteration,
which aligns with the functions `pairwise` and `windowed` that deal with
iterating over elements in a collection.

Let's consider the reasoning for selecting the files:

1. **boltons/iterutils.py**: This file is the most likely candidate as it
 is named to suggest it contains utilities for iteration, which is
directly related to the `pairwise` and `windowed` functions mentioned in
the problem.

2. **boltons/funcutils.py**: This file might contain functional utilities
, which could include higher-order functions or iterators. If `pairwise`
and `windowed` are implemented using functional programming paradigms,
they might be here.

3. **boltons/listutils.py**: This file could potentially contain
utilities for list operations. If `pairwise` and `windowed` are
specifically designed for lists, they might be implemented here.

4. **boltons/typeutils.py**: This file might contain utilities related to
 type handling, which could include generic functions for handling
iterables of various types.

5. **boltons/strutils.py**: Although less likely, if `pairwise` and `
windowed` have any specific implementations for strings or string-like
iterables, they might be here.
```

```
Now, let's finalize the top 5 files based on the reasoning:

```
boltons/iterutils.py
boltons/funcutils.py
boltons/listutils.py
boltons/typeutils.py
boltons/strutils.py
```

>>>>>>>>>>>>>>>>>>> user:

Please look through the following problem description and repository
structure and provide a list of files that one would need to edit to
solve the software development problem.
I have already find 5 relevant files. Accrording to the import relations,
 construct the call graph first.

### Problem Description ###
<request>
Adding fill parameters to pairwise and windowed
I found these handy for when I needed lookahead/lookbehind, but didn't
want to add extra processing code before/after the processing loop. Here'
s a sketch of one way I've used it:

```python
for record, next_record in pairwise_iter(records, fill=None):
  if next_record is not None:
      record.link(next_record)
  record.process()
```

Without fill, I have to call `records[-1].process()` after the loop,
assuming it's a list and not an iterable.
</request>

###

### Repository Structure ###
boltons/
    __init__.py
    cacheutils.py
    debugutils.py
    deprutils.py
    dictutils.py
    easterutils.py
    ecoutils.py
    excutils.py
    fileutils.py
    formatutils.py
    funcutils.py
    gcutils.py
    ioutils.py
    iterutils.py
    jsonutils.py
    listutils.py
    mathutils.py
    mboxutils.py
    namedutils.py
    pathutils.py
    queueutils.py
    setutils.py
    socketutils.py
    statsutils.py
```

```
      strutils.py
      tableutils.py
      tbutils.py
      timeutils.py
      typeutils.py
      urlutils.py
docs/
      conf.py
misc/
      bench_omd.py
      linkify_changelog.py
      table_html_app.py
setup.py

###

### Files To Be Explored ###
['boltons/iterutils.py', 'boltons/funcutils.py', 'boltons/listutils.py',
'boltons/typeutils.py', 'boltons/strutils.py']

###

### Import Relations ###
file: boltons/iterutils.py
 ['import os', 'import math', 'import time', 'import codecs', 'import
 random', 'import itertools']

file: boltons/funcutils.py
 ['from __future__ import print_function', 'import sys', 'import re', '
 import inspect', 'import functools', 'import itertools', 'from types
 import MethodType, FunctionType']

file: boltons/listutils.py
 ['from __future__ import print_function, division', 'import operator', '
 from math import log as math_log', 'from itertools import chain, islice
 ']

file: boltons/typeutils.py
 ['import sys', 'from collections import deque']

file: boltons/strutils.py
 ['from __future__ import print_function', 'import re', 'import sys', '
 import uuid', 'import zlib', 'import string', 'import unicodedata', '
 import collections', 'from gzip import GzipFile']

###

Based on the import relationships, please analyze which files in the
repository depend on which other files within the same repository. Ignore
 any libraries or modules that are imported from outside the current
repository. Present the results in the following format:

file1.py -> file6.py, file7.py
file2.py ->
file3.py -> xxx/file8.py
file4.py -> file2.py, file9.py
file5.py -> xxx/file10.py, xxx/file11.py

Note: Solving the problem not only requires determining where to modify
the code, but also identifying which other code to refer to in order to
understand and invoke.

Based on the files listed above and the import relations, reconfirm and
provide the complete names of the top 10 most likely relevant files,
```

considering both where changes need to be made and which files are
important to refer to.

Please think step by step and give you reasoning process first. Finally,
provide full path and return top 10 files.

The final returned files should be separated by new lines ordered by most
 to least important and wrapped with ```
For example:
```
file1.py
file2.py
file3.py
file4.py
file5.py
file6.py
file7.py
file8.py
file9.py
file10.py
```
Note: file1.py indicates the top-1 file, file2.py indicates the top-2
file, and so on. Do not include test files.

>>>>>>>>>>>>>>>>>>>> assistant:
To solve the problem, we need to analyze the import relationships and
determine which files are relevant for adding the "fill" parameter to the
 `pairwise` and `windowed` functions. Here's the step-by-step reasoning
process:

1. **Identify the Core Functionality**: The problem is about adding a "
fill" parameter to the `pairwise` and `windowed` functions. These
functions are likely to be part of the `iterutils.py` file, as they deal
with iteration utilities.

2. **Analyze Import Relations**: We need to understand the dependencies
between the files to see if any other files might be relevant for
understanding or modifying the `pairwise` and `windowed` functions.

3. **Check Import Statements**:
   - `boltons/iterutils.py` imports standard libraries like `os`, `math`,
     `time`, `codecs`, `random`, and `itertools`. It does not import any
   other files from the `boltons` package.
   - `boltons/funcutils.py` imports `itertools` and other standard
   libraries, but does not import `iterutils`.
   - `boltons/listutils.py` imports `itertools` and other standard
   libraries, but does not import `iterutils`.
   - `boltons/typeutils.py` and `boltons/strutils.py` do not import `
   iterutils` or any other relevant files.

4. **Determine Relevant Files**: Since `iterutils.py` does not import or
is not imported by any other files in the list, it is likely self-
contained for the functionality of `pairwise` and `windowed`. However,
understanding how these functions might interact with other utilities
could be useful, so we consider the files that import `itertools` as they
 might have similar iteration-related utilities.

5. **Prioritize Files**: Based on the above analysis, `iterutils.py` is
the most relevant file for direct modification. The other files (`
funcutils.py`, `listutils.py`, `typeutils.py`, `strutils.py`) might
provide additional context or similar implementations that could be
useful for reference.

```
6. **Top 10 Files**: Since the problem is specific to iteration utilities
, the most relevant file is 'iterutils.py'. The other files are included
for reference and understanding of similar patterns or utilities.

Based on this reasoning, here are the top 10 files, ordered by relevance:

```
boltons/iterutils.py
boltons/funcutils.py
boltons/listutils.py
boltons/typeutils.py
boltons/strutils.py
boltons/cacheutils.py
boltons/debugutils.py
boltons/deprutils.py
boltons/dictutils.py
boltons/easterutils.py
```

Note: The last five files are included to fill the top 10 list, assuming
they might provide additional context or utility functions that could be
relevant indirectly.

>>>>>>>>>>>>>>>>>>>> user:

Please look through the following problem description and repository
structure and provide a list of files that one would need to edit to
solve the software development problem.
I have already find 10 relevent files according to the file structure and
 dependency.
I will further give the internal structures of these 10 files.
Please rank them again and reflect the result according to the internal
structures and dependency.

### Problem Description ###
<request>
Adding fill parameters to pairwise and windowed
I found these handy for when I needed lookahead/lookbehind, but didn't
want to add extra processing code before/after the processing loop. Here'
s a sketch of one way I've used it:

```python
for record, next_record in pairwise_iter(records, fill=None):
  if next_record is not None:
      record.link(next_record)
  record.process()
```

Without fill, I have to call 'records[-1].process()' after the loop,
assuming it's a list and not an iterable.
</request>

###

### Import Relations ###
file: boltons/iterutils.py
 ['import os', 'import math', 'import time', 'import codecs', 'import
 random', 'import itertools']

file: boltons/funcutils.py
 ['from __future__ import print_function', 'import sys', 'import re', '
 import inspect', 'import functools', 'import itertools', 'from types
 import MethodType, FunctionType']

file: boltons/listutils.py
```

```
1458    ['from __future__ import print_function, division', 'import operator', '
1459     from math import log as math_log', 'from itertools import chain, islice
1460    ']

1461

1462    file: boltons/typeutils.py
1463     ['import sys', 'from collections import deque']

1464    file: boltons/strutils.py
1465     ['from __future__ import print_function', 'import re', 'import sys', '
1466     import uuid', 'import zlib', 'import string', 'import unicodedata', '
1467     import collections', 'from gzip import GzipFile']

1468

1469    ###

1470

1471

1472    ### Files To Be Ranked ###
1473    file: boltons/iterutils.py
1474            class: ['PathAccessError', 'GUIDerator', 'SequentialGUIDerator']
1475            static functions:  ['is_iterable', 'is_scalar', 'is_collection',
                'split', 'split_iter', 'lstrip', 'lstrip_iter', 'rstrip', '
1476            rstrip_iter', 'strip', 'strip_iter', 'chunked', '
1477            _validate_positive_int', 'chunked_iter', 'chunk_ranges', '
1478            pairwise', 'pairwise_iter', 'windowed', 'windowed_iter', 'xfrange
1479            ', 'frange', 'backoff', 'backoff_iter', 'bucketize', 'partition',
                 'unique', 'unique_iter', 'redundant', 'one', 'first', '
1480            flatten_iter', 'flatten', 'same', 'default_visit', 'default_enter
1481            ', 'default_exit', 'remap', 'get_path', 'research', 'soft_sorted
1482            ', 'untyped_sorted']
1483            class functions: [
                    PathAccessError: ['__init__', '__repr__', '__str__']
1484                GUIDerator: ['__init__', 'reseed', '__iter__', '__next__
1485                ', '__next__']
1486                SequentialGUIDerator: ['reseed', 'reseed', '__next__']
1487            ]
1488    file: boltons/funcutils.py
1489            class: ['InstancePartial', 'CachedInstancePartial', '
                FunctionBuilder', 'MissingArgument', 'ExistingArgument']
1490            static functions:  ['inspect_formatargspec', '
1491            get_module_callables', 'mro_items', 'dir_dict', 'copy_function',
1492            'partial_ordering', 'format_invocation', 'format_exp_repr', '
                format_nonexp_repr', 'wraps', 'update_wrapper', '
1493            _parse_wraps_expected', '_indent', 'total_ordering', 'noop']
1494            class functions: [
1495                InstancePartial: ['_partialmethod', '__get__']
1496                CachedInstancePartial: ['_partialmethod', '__set_name__',
                    '__get__']
1497                FunctionBuilder: ['_argspec_to_dict', '_argspec_to_dict',
1498                    '__init__', 'get_sig_str', 'get_invocation_str', '
1499                get_sig_str', 'get_invocation_str', 'from_func', '
1500                get_func', 'get_defaults_dict', 'get_arg_names', 'add_arg
1501                ', 'add_arg', 'remove_arg', '_compile']
1502                MissingArgument: []
1503                ExistingArgument: []
            ]
1504    file: boltons/listutils.py
1505            class: ['BarrelList', 'SplayList']
1506            static functions:  []
1507            class functions: [
1508                BarrelList: ['__init__', '_cur_size_limit', '
1509                _translate_index', '_balance_list', 'insert', 'append', '
1510                extend', 'pop', 'iter_slice', 'del_slice', 'from_iterable
1511                ', '__iter__', '__reversed__', '__len__', '__contains__',
                    '__getitem__', '__delitem__', '__setitem__', '
```

```
                    __getslice__', '__setslice__', '__repr__', 'sort', '
                    reverse', 'count', 'index']
                SplayList: ['shift', 'swap']
        ]
file: boltons/typeutils.py
        class: ['classproperty']
        static functions:  ['make_sentinel', 'issubclass', '
        get_all_subclasses']
        class functions: [
                classproperty: ['__init__', '__get__']
        ]
file: boltons/strutils.py
        class: ['DeaccenterDict', 'HTMLTextExtractor', 'MultiReplace']
        static functions:  ['camel2under', 'under2camel', 'slugify', '
        split_punct_ws', 'unit_len', 'ordinalize', 'cardinalize', '
        singularize', 'pluralize', '_match_case', 'find_hashtags', 'a10n
        ', 'strip_ansi', 'asciify', 'is_ascii', 'bytes2human', 'html2text
        ', 'gunzip_bytes', 'gzip_bytes', 'iter_splitlines', 'indent', '
        is_uuid', 'escape_shell_args', 'args2sh', 'args2cmd', '
        parse_int_list', 'format_int_list', 'complement_int_list', '
        int_ranges_from_int_list', 'multi_replace', 'unwrap_text']
        class functions: [
                DeaccenterDict: ['__missing__', '__getitem__']
                HTMLTextExtractor: ['__init__', 'handle_data', '
                handle_charref', 'handle_entityref', 'get_text']
                MultiReplace: ['__init__', '_get_value', 'sub']
        ]
file: boltons/cacheutils.py
        class: ['RLock', 'LRI', 'LRU', '_HashedKey', 'CachedFunction', '
        CachedMethod', 'cachedproperty', 'ThresholdCounter', 'MinIDMap']
        static functions:  ['make_cache_key', 'cached', 'cachedmethod']
        class functions: [
                RLock: ['__enter__', '__exit__']
                LRI: ['__init__', '_init_ll', '_print_ll', '
                _get_flattened_ll', '_get_link_and_move_to_front_of_ll',
                '_set_key_and_add_to_front_of_ll', '
                _set_key_and_evict_last_in_ll', '_remove_from_ll', '
                __setitem__', '__getitem__', 'get', '__delitem__', 'pop',
                 'popitem', 'clear', 'copy', 'setdefault', 'update', '
                __eq__', '__ne__', '__repr__']
                LRU: ['__getitem__']
                _HashedKey: ['__init__', '__hash__', '__repr__']
                CachedFunction: ['__init__', '__call__', '__repr__']
                CachedMethod: ['__init__', '__get__', '__call__', '
                __repr__']
                cachedproperty: ['__init__', '__get__', '__repr__']
                ThresholdCounter: ['__init__', 'threshold', 'add', '
                elements', 'most_common', 'get_common_count', '
                get_uncommon_count', 'get_commonality', '__getitem__', '
                __len__', '__contains__', 'iterkeys', 'keys', 'itervalues
                ', 'values', 'iteritems', 'items', 'get', 'update']
                MinIDMap: ['__init__', 'get', 'drop', '_clean', '
                __contains__', '__iter__', '__len__', 'iteritems']
        ]
file: boltons/debugutils.py
        class: []
        static functions:  ['pdb_on_signal', 'pdb_on_exception', '
        trace_print_hook', 'wrap_trace']
        class functions: [
        ]
file: boltons/deprutils.py
        class: ['DeprecatableModule']
        static functions:  ['deprecate_module_member']
        class functions: [
                DeprecatableModule: ['__init__', '__getattribute__']
```

```
            ]
file: boltons/dictutils.py
        class: ['OrderedMultiDict', 'FastIterOrderedMultiDict', 'OneToOne
        ', 'ManyToMany', 'FrozenHashError', 'FrozenDict']
        static functions:  ['subdict']
        class functions: [
                OrderedMultiDict: ['__new__', '__init__', '__getstate__',
                 '__setstate__', '_clear_ll', '_insert', 'add', 'addlist
                ', 'get', 'getlist', 'clear', 'setdefault', 'copy', '
                fromkeys', 'update', 'update_extend', '__setitem__', '
                __getitem__', '__delitem__', '__eq__', '__ne__', '__ior__
                ', 'pop', 'popall', 'poplast', '_remove', '_remove_all',
                'iteritems', 'iterkeys', 'itervalues', 'todict', 'sorted
                ', 'sortedvalues', 'inverted', 'counts', 'keys', 'values
                ', 'items', '__iter__', '__reversed__', '__repr__', '
                viewkeys', 'viewvalues', 'viewitems']
                FastIterOrderedMultiDict: ['_clear_ll', '_insert', '
                _remove', '_remove_all', 'iteritems', 'iterkeys', '
                __reversed__']
                OneToOne: ['__init__', 'unique', '__setitem__', '
                __delitem__', 'clear', 'copy', 'pop', 'popitem', '
                setdefault', 'update', '__repr__']
                ManyToMany: ['__init__', 'get', '__getitem__', '
                __setitem__', '__delitem__', 'update', 'add', 'remove', '
                replace', 'iteritems', 'keys', '__contains__', '__iter__
                ', '__len__', '__eq__', '__repr__']
                FrozenHashError: []
                FrozenDict: ['updated', 'fromkeys', '__repr__', '
                __reduce_ex__', '__hash__', '__copy__', '
                _raise_frozen_typeerror']
        ]
file: boltons/easterutils.py
        class: []
        static functions:  ['gobs_program']
        class functions: [
        ]

###

Please think step by step and give you reasoning process first. Finally,
provide full path and return top 5 files.

The returned files should be separated by new lines ordered by most to
least important and wrapped with '''
For example:
'''
file1.py
file2.py
file3.py
file4.py
file5.py
'''
Note: file1.py indicates the top-1 file, file2.py indicates the top-2
file, and so on. Do not include test files.

>>>>>>>>>>>>>>>>>>>> assistant:
To solve the problem of adding fill parameters to the 'pairwise' and '
windowed' functions, we need to identify the files that contain these
functions and any related utility functions that might be affected by or
need to support the new feature.

### Step-by-Step Reasoning:
```

```
1. **Identify the Functions:**
   - The problem description specifically mentions `pairwise_iter` and
   implies a similar change might be needed for `windowed` functions. We
   need to locate these functions in the codebase.

2. **Locate the Functions:**
   - From the provided internal structure of the files, we see that `
   pairwise`, `pairwise_iter`, `windowed`, and `windowed_iter` are static
    functions in `boltons/iterutils.py`.

3. **Assess Dependencies:**
   - Since these functions are in `boltons/iterutils.py`, this file is
   the primary candidate for modification.
   - We should also consider if any other files might be using these
   functions or if there are utility functions that might need to be
   updated to support the new `fill` parameter.

4. **Check for Related Utilities:**
   - The `boltons/iterutils.py` file might have other utility functions
   that interact with `pairwise` and `windowed`. We should ensure that
   any such functions are also updated if necessary.

5. **Consider Other Files:**
   - While other files like `boltons/funcutils.py`, `boltons/listutils.py
   `, etc., are part of the repository, they do not contain the functions
    in question or directly relate to the problem at hand based on the
   provided information.

6. **Rank the Files:**
   - Given the focus on `pairwise` and `windowed`, `boltons/iterutils.py`
    is the most relevant file.
   - Other files are less relevant unless they are found to directly use
   or extend the functionality of `pairwise` and `windowed`.

### Conclusion:

Based on the above reasoning, the top file to edit is `boltons/iterutils.
py`. Other files are not directly relevant to the problem as described,
so they are not included in the top 5 list.

```
boltons/iterutils.py
```
```

## F.2 COMPONENT LOCALIZATION

```
>>>>>>>>>>>>>>>>>>>> system:

You will be presented with a repository-level coding problem with
repository file structure to access the source code of the software.
Since the modification is based on the code repository, the modified
locations may include files, classes, and functions, and the
modifications may be in the form of addition, deletion, or update.
Your task is to locate the top-5 most likely edit locations based on the
problem description and the information you retrieve using given
functions.

Function calls you can use are as follows:
* get_code_of_class('file_name', 'class_name') -> Get the code of a
specified class in the given file and python project. 'file_name' -> The
name of the file. 'class_name' -> The name of the class. *
* get_code_of_class_function('file_name', 'class_name', 'func_name') ->
Get the code of a specified function in the given class, file, and python
```

```
1674    project. 'file_name' -> The name of the file. 'class_name' -> The name
1675    of the class. 'func_name' -> The name of the function. *
1676    * get_code_of_file_function('file_name', 'func_name') -> Get the code of
1677    a specified function in the given file and python project. 'file_name' ->
1678     The name of the file. 'func_name' -> The name of the function. *
1679    * get_toplevel_code('file_name') -> Get all the code in a given file that
1680     is not part of a class or function definition. This is useful for
1681    viewing imports, global variables, constants, and any top-level script
1682    logic. 'file_name' -> The name of the file. *
1683    * exit() -> Exit function calling to give your final answer when you are
1684    confident of the answer. *
1685
1686    You have 10 chances to call function.
1687
1688
1689    >>>>>>>>>>>>>>>>>>>> user:
1690    ### Problem Description ###
1691    <request>
1692    (PUP-5548) Use 'service' in debian provider to interact with all init
1693    systems
1694    In the Debian family, the 'service' command provides an
1695    abstraction around available init systems, allowing it to be used
1696    to properly determine service state regardless of which init system
1697    is active on the system.
1698
1699    This commit updates the Debian provider's 'statuscmd' method to use
1700    'service' at all times, allowing the provider to properly query
1701    services regardless of OS version or init system in place.
1702    </request>
1703
1704    ###
1705
1706    Let's locate the relevant elements (function/class) step by step using
1707    reasoning and function calls.
1708    I have pre-identified top-5 relevant files. There stuctures are as
1709    follows:
1710    file: lib/puppet/provider/service/debian.rb
1711            class: []
1712            static functions:  ['disable', 'enabled?', 'get_start_link_count
1713            ', 'enable', 'statuscmd']
1714            class functions: [
1715            ]
1716    file: lib/puppet/provider/service/base.rb
1717            class: []
1718            static functions:  ['getps', 'getpid', 'status', 'statuscmd', '
1719            start', 'startcmd', 'stop', 'stopcmd']
1720            class functions: [
1721            ]
1722    file: lib/puppet/type/service.rb
1723            class: ['Puppet']
1724            static functions:  []
1725            class functions: [
1726                    Puppet: ['retrieve', 'retrieve', 'sync', 'refresh']
1727            ]
        file: lib/puppet/provider/service/systemd.rb
                class: []
                static functions:  ['instances', 'disable', 'get_start_link_count
                ', 'enabled?', 'debian_enabled?', 'status', 'enable', 'mask', '
                unmask', 'restartcmd', 'startcmd', 'stopcmd']
                class functions: [
                ]
        file: lib/puppet/provider/service/init.rb
                class: []
                static functions:  ['defpath', 'excludes', 'instances', '
                get_services', 'hasstatus=', 'initscript', 'paths', 'search', '
```

```
1728            startcmd', 'stopcmd', 'restartcmd', 'texecute', 'statuscmd', '
1729            is_init?']
1730            class functions: [
1731            ]
1732    The formal parameter 'file_name' takes the value in "file:"
        The formal parameter 'class_name' takes the value in "class:"
1733    The formal parameter 'func_name' takes the value in "static functions:"
1734     and "class functions: "
        Avoid making multiple identical calls to save overhead.
1735    You must strictly follow the structure I give to call different tools.
1736    For static functions, you can use 'get_code_of_file_function', and for
1737     class functions, you can use 'get_code_of_class_function'.
        In order to locate accurately, you can pre-select 7 locations, then check
1738     them through function calls, and finally confirm 5 file names.
1739    Don't make the first function call in this message.
1740
1741
1742
1743    >>>>>>>>>>>>>>>>>>>> assistant:
        To address the problem, we need to focus on the 'statuscmd' method in the
1744     Debian provider, as the request specifically mentions updating this
1745    method to use the 'service' command. Based on the provided file structure
1746    , the relevant files and functions to consider are:
1747
1748    1. **lib/puppet/provider/service/debian.rb**
1749        - Function: 'statuscmd'
1750    2. **lib/puppet/provider/service/base.rb**
1751        - Function: 'statuscmd'
1752
1753    3. **lib/puppet/provider/service/systemd.rb**
1754        - Function: 'statuscmd'
1755
1756    4. **lib/puppet/provider/service/init.rb**
        - Function: 'statuscmd'
1757
1758    These files contain the 'statuscmd' function, which is likely related to
        the service status command functionality. The problem description
1759    specifically mentions the Debian provider, so the 'debian.rb' file is of
1760    primary interest. However, examining the 'statuscmd' function in other
1761    related files might provide additional context or necessary changes.
1762
1763    Let's proceed with checking the code of these functions to identify the
        necessary modifications. I will start by examining the 'statuscmd'
1764    function in 'lib/puppet/provider/service/debian.rb'.
1765
1766    >>>>>>>>>>>>>>>>>>>> user:
1767
1768    Now call a function in this format 'FunctionName(Argument)' in a single
        line without any other word or signal (such as ```).
1769    Don't call the same function you've previous called, because this may
1770    waste your context length.
1771
1772
1773    >>>>>>>>>>>>>>>>>>>> assistant:
        get_code_of_file_function('lib/puppet/provider/service/debian.rb', '
1774    statuscmd')
1775
1776    >>>>>>>>>>>>>>>>>>>> user:
1777    I have already checked this function/class is related to the task.
1778
1779    >>>>>>>>>>>>>>>>>>>> user:
          def statuscmd
1780        os = Facter.value(:operatingsystem).downcase
1781
            if os == 'debian'
```

```
      majversion = Facter.value(:operatingsystemmajrelease).to_i
    else
      majversion = Facter.value(:operatingsystemmajrelease).split('.')
      [0].to_i
    end

    if ((os == 'debian' && majversion >= 8) || (os == 'ubuntu' &&
    majversion >= 15))
      # SysVInit scripts will always return '0' for status when the
      service is masked,
      # even if the service is actually stopped. Use the SysVInit-Systemd
       compatibility
      # layer to determine the actual status. This is only necessary when
       the SysVInit
      # version of a service is queried. I.e, 'ntp' instead of 'ntp.
      service'.
      (@resource[:hasstatus] == :true) && ["systemctl", "is-active",
      @resource[:name]]
    else
      super
    end
  end

>>>>>>>>>>>>>>>>>>>>> user:

Now call a function in this format 'FunctionName(Argument)' in a single
line without any other word or signal (such as ```).
Don't call the same function you've previous called, because this may
waste your context length.

You can check the functions that this part of code calls.

>>>>>>>>>>>>>>>>>>>>> assistant:
get_code_of_file_function('lib/puppet/provider/service/base.rb', '
statuscmd')

>>>>>>>>>>>>>>>>>>>>> user:
I have already checked this function/class is related to the task.

>>>>>>>>>>>>>>>>>>>>> user:
  def statuscmd
  end

>>>>>>>>>>>>>>>>>>>>> user:

Now call a function in this format 'FunctionName(Argument)' in a single
line without any other word or signal (such as ```).
Don't call the same function you've previous called, because this may
waste your context length.

You can check the functions that this part of code calls.

>>>>>>>>>>>>>>>>>>>>> assistant:
get_code_of_file_function('lib/puppet/provider/service/systemd.rb', '
statuscmd')

>>>>>>>>>>>>>>>>>>>>> user:
The call of get_code_of_file_function('lib/puppet/provider/service/
systemd.rb', 'statuscmd') encountered error. Maybe you retrieved a
location which does not exist.

>>>>>>>>>>>>>>>>>>>>> user:
```

```
Now call a function in this format 'FunctionName(Argument)' in a single
line without any other word or signal (such as ```).
Don't call the same function you've previous called, because this may
waste your context length.

>>>>>>>>>>>>>>>>>>>> assistant:
get_code_of_file_function('lib/puppet/provider/service/init.rb', '
statuscmd')

>>>>>>>>>>>>>>>>>>>> user:
I have already checked this function/class is related to the task.

>>>>>>>>>>>>>>>>>>>> user:
  def statuscmd
    (@resource[:hasstatus] == :true) && [initscript, :status]
  end

>>>>>>>>>>>>>>>>>>>> user:

Now call a function in this format 'FunctionName(Argument)' in a single
line without any other word or signal (such as ```).
Don't call the same function you've previous called, because this may
waste your context length.

You can check the functions that this part of code calls.

>>>>>>>>>>>>>>>>>>>> assistant:
get_code_of_file_function('lib/puppet/provider/service/init.rb', '
initscript')

>>>>>>>>>>>>>>>>>>>> user:
I have already checked this function/class is not related to the task.

>>>>>>>>>>>>>>>>>>>> user:
  def initscript
    @initscript ||= self.search(@resource[:name])
  end

>>>>>>>>>>>>>>>>>>>> user:

Now call a function in this format 'FunctionName(Argument)' in a single
line without any other word or signal (such as ```).
Don't call the same function you've previous called, because this may
waste your context length.

Don't check the functions that this part of code calls.

>>>>>>>>>>>>>>>>>>>> assistant:
get_toplevel_code(lib/puppet/provider/service/debian.rb)

>>>>>>>>>>>>>>>>>>>> user:
I have already checked this function/class is related to the task.

>>>>>>>>>>>>>>>>>>>> user:
# Manage debian services.  Start/stop is the same as InitSvc, but enable/
disable
# is special.
Puppet::Type.type(:service).provide :debian, :parent => :init do
  desc <<-EOT
    Debian's form of `init`-style management.
```

```
     The only differences from `init` are support for enabling and
     disabling
     services via `update-rc.d` and the ability to determine enabled
     status via
     `invoke-rc.d`.

   EOT

   commands :update_rc => "/usr/sbin/update-rc.d"
   # note this isn't being used as a command until
   # http://projects.reductivelabs.com/issues/2538
   # is resolved.
   commands :invoke_rc => "/usr/sbin/invoke-rc.d"

   defaultfor :operatingsystem => :cumuluslinux
   defaultfor :operatingsystem => :debian, :operatingsystemmajrelease =>
   ['5','6','7']

   # Remove the symlinks
...
end
...

>>>>>>>>>>>>>>>>>>>> user:

Now call a function in this format 'FunctionName(Argument)' in a single
line without any other word or signal (such as ```).
Don't call the same function you've previous called, because this may
waste your context length.

You can check the functions that this part of code calls.

>>>>>>>>>>>>>>>>>>>> assistant:
exit()

>>>>>>>>>>>>>>>>>>>> user:

Based on the available information, reconfirm and provide complete names
of the most likely edit locations (10 locations at most).
Before make the final decision, please check whether the function name is
 correct or not, for static functions, don't add class name.
file: lib/puppet/provider/service/debian.rb
        class: []
        static functions:  ['disable', 'enabled?', 'get_start_link_count
        ', 'enable', 'statuscmd']
        class functions: [
        ]
file: lib/puppet/provider/service/base.rb
        class: []
        static functions:  ['getps', 'getpid', 'status', 'statuscmd', '
        start', 'startcmd', 'stop', 'stopcmd']
        class functions: [
        ]
file: lib/puppet/type/service.rb
        class: ['Puppet']
        static functions:  []
        class functions: [
                Puppet: ['retrieve', 'retrieve', 'sync', 'refresh']
        ]
file: lib/puppet/provider/service/systemd.rb
        class: []
        static functions:  ['instances', 'disable', 'get_start_link_count
        ', 'enabled?', 'debian_enabled?', 'status', 'enable', 'mask', '
        unmask', 'restartcmd', 'startcmd', 'stopcmd']
```

```
        class functions: [
        ]
file: lib/puppet/provider/service/init.rb
        class: []
        static functions:  ['defpath', 'excludes', 'instances', '
        get_services', 'hasstatus=', 'initscript', 'paths', 'search', '
        startcmd', 'stopcmd', 'restartcmd', 'texecute', 'statuscmd', '
        is_init?']
        class functions: [
        ]

Please provide the complete set of locations as either a class name, a
function name, or a file name.
The returned files should be separated by new lines ordered by most to
least important and wrapped with ```
Since your answer will be processed automatically, please give your
answer in the exapmle format as follows.
```
top1_file_fullpath.py
function: Class1.Function1

top2_file_fullpath.py
function: Function2

top3_file_fullpath.py
class: Class3

top4_file_fullpath.py
function: Class4.Function4

top5_file_fullpath.py
function: Function5

top6_file_fullpath.py
global

top7_file_fullpath.py
function: Class7.Function7
```
Replace the 'Top_file_fullpath.py' with the actual file path, the 'Class'
 with the actual class name and the 'Function' with the actual function
name. 'global' means the code is not in a class or function.
For example,
```
sklearn/linear_model/__init__.py
function: LinearRegression.fit
```

>>>>>>>>>>>>>>>>>>> assistant:
```
lib/puppet/provider/service/debian.rb
global

lib/puppet/provider/service/debian.rb
function: statuscmd

lib/puppet/provider/service/base.rb
function: statuscmd

lib/puppet/provider/service/init.rb
function: statuscmd
```
```

### F.3   CODE EDITING

```
>>>>>>>>>>>>>>>>>>>> user:

We are currently solving the following task within our repository. Here
is the task description.

### Task Description ###
<request>
Add port mapping on run
CF issue : https://github.com/jolicode/docker-starter/issues/31
</request>

<discussion>
-------------------
Mapping ports on "run" command
Hi there,

Is there any reason to avoid mapping ports on "run" command ?
https://github.com/jolicode/docker-starter/blob/
c319756cc1ab1de8e4a3f29bea306695752f84c2/fabfile.py#L179

I just needed to add --service-ports option to the fabfile.py to run my
container with port mapping embed.

:kissing_heart:

**********
since we use docker-compose, it's better to set the port mapping in it.
But if you need something special, you can use port mapping via the run
command.

Feel free to modify your fabfile. This docker-starter is not a lib, just
a starter that must be customized according to your needs.
--service-ports does actually the mapping defined in docker-compose.yaml.
**********

</discussion>

###

Below are some code segments, each from a relevant file. One or more of
these files may need to be edited to solve the task.

--- BEGIN FILE ---
```
### fabfile.py
...
env.extra_domains = []
# This is the host directory containing your PHP application
env.project_directory = 'application'

services_to_build_first = [
    'php-base',
    'builder',
]

def with_builder(func):
    @wraps(func)
    def decorated(*args, **kwargs):
        compose_files = env.compose_files[:]
```

```
            env.compose_files = ['docker-compose.builder.yml'] + env.
        compose_files
            ret = func(*args, **kwargs)
            env.compose_files = compose_files

            return ret
        return decorated

    @with_builder
    def build():
        """
        Build the infrastructure
        """
        command = 'build'
        command += ' --build-arg PROJECT_NAME=%s' % env.project_name
        command += ' --build-arg USER_ID=%s' % env.user_id

        for service in services_to_build_first:
            commandForService = '%s %s' % (command, service)
            docker_compose(commandForService)

        docker_compose(command)

    @task
    def up():
        """
        Build and start the infrastructure
        """
        build()
        docker_compose('up --remove-orphans -d')

    ...

    @task
    @with_builder
    def destroy():
        """
        Clean the infrastructure (remove container, volume, networks)
        """
        docker_compose('down --volumes --rmi=local')

    def run_in_docker_or_locally_for_dinghy(command):
        """
        Mac users have a lot of problems running Yarn / Webpack on the Docker
         stack so this func allow them to run these tools on their host
        """
        if env.dinghy:
            local('cd %s && %s' % (env.project_directory, command))
        else:
            docker_compose_run(command)

    def docker_compose(command_name):

        domains = '`' + '`, `'.join([env.root_domain] + env.extra_domains) +
        '`'

        localEnv = {
            'PROJECT_NAME': env.project_name,
            'PROJECT_DIRECTORY': env.project_directory,
            'PROJECT_ROOT_DOMAIN': env.root_domain,
            'PROJECT_DOMAINS': domains,
```

```
      }

    with shell_env(**localEnv):
        local('docker-compose -p %s %s %s' % (
            env.project_name,
            ' '.join('-f ' + env.root_dir + '/infrastructure/docker/' +
            file for file in env.compose_files),
            command_name
        ))

def docker_compose_run(command_name, service="builder", user="app",
no_deps=False, workdir=None):
    args = [
        'run ',
        '--rm ',
        '-u %s ' % _shell_escape(user),
    ]

    if no_deps:
        args.append('--no-deps ')

    if workdir is not None:
        args.append('-w %s ' % _shell_escape(workdir))

    docker_compose('%s %s /bin/bash -c "exec %s"' % (
        ' '.join(args),
        _shell_escape(service),
        _shell_escape(command_name)
    ))

def set_local_configuration():
    env.compose_files = ['docker-compose.yml']
    env.user_id = int(local('id -u', capture=True))
    env.root_dir = os.path.dirname(os.path.abspath(__file__))

    if env.user_id > 256000:
        env.user_id = 1000

...

```
--- END FILE ---

Please first localize the positions to edit based on the task statement,
and then output the files that need to be deleted, modified or added.
'- file' means deleting the file;
'* file' means modificating the file;
'+ file' means adding the file.
The file should list like below:
```
- file1.py
* file2.py
* file3.py
+ file4.py
```

To solve the task, you should then generate *SEARCH/REPLACE* edits.

Every *SEARCH/REPLACE* edit must use this format:
1. The file path
2. The start of search block: <<<<<<< SEARCH

```
3. A contiguous chunk of lines to search for in the existing source code
4. The dividing line: =======
5. The lines to replace into the source code
6. The end of the replace block: >>>>>>> REPLACE

Here is an example:

```python
### mathweb/flask/app.py
<<<<<<< SEARCH
from flask import Flask
=======
import math
from flask import Flask
>>>>>>> REPLACE
```

Please note that the *SEARCH/REPLACE* edit REQUIRES PROPER INDENTATION.
If you would like to add the line '          print(x)', you must fully
write that out, with all those spaces before the code!
Wrap the *SEARCH/REPLACE* edit in blocks ```python...```.
When multiple edits should be done, please output *SEARCH/REPLACE* edit
one by one and give your reasoning process before each *SEARCH/REPLACE*
block.

>>>>>>>>>>>>>>>>>>>> assistant:
The plan of solving this software task:
To address the task of adding port mapping on the "run" command, we need
to modify the `docker_compose_run` function in the `fabfile.py`. The
discussion suggests using the `--service-ports` option to enable port
mapping as defined in the `docker-compose.yml` file. We will add a
parameter to the `docker_compose_run` function to allow this option to be
 toggled.

The files that should be edited include:
* fabfile.py

Here are my edits for code.

The motivation of this edit:
We need to modify the `docker_compose_run` function to include an
optional parameter for port mapping. By adding a `port_mapping` parameter
, we can control whether the `--service-ports` option is included in the
Docker run command. This change allows users to enable port mapping when
running a service, which is useful for exposing ports as defined in the `
docker-compose.yml` file.

```python
### fabfile.py
<<<<<<< SEARCH
        ))

def docker_compose_run(command_name, service="builder", user="app",
no_deps=False, workdir=None):
    args = [
        'run ',
        '--rm ',
=======
        ))

def docker_compose_run(command_name, service="builder", user="app",
no_deps=False, workdir=None, port_mapping=False):
```

```
    args = [
        'run ',
        '--rm ',
>>>>>>> REPLACE
<<<<<<< SEARCH
    if no_deps:
        args.append('--no-deps ')

    if workdir is not None:
        args.append('-w %s ' % _shell_escape(workdir))

=======
    if no_deps:
        args.append('--no-deps ')

    if port_mapping:
        args.append('--service-ports ')

    if workdir is not None:
        args.append('-w %s ' % _shell_escape(workdir))

>>>>>>> REPLACE
```

### F.4 CODE COMPLETION

```
>>>>>>>>>>>>>>>>>>>> user:

We are currently solving the following task within our repository. Here
is the task description.

### Task Description ###
Please help me to complete the code at the position of '[TODO]', in the
b2/sync/scan_policies.py
    def __init__(
        self,
        exclude_dir_regexes=tuple(),
        exclude_file_regexes=tuple(),
        include_file_regexes=tuple(),
    ):

###

Below are some code segments, each from a relevant file. One or more of
these files may need to be edited to solve the task.

--- BEGIN FILE ---
```
### b2/sync/scan_policies.py
######################################################################
#
# File: b2/sync/scan_policies.py
#
# Copyright 2018 Backblaze Inc. All Rights Reserved.
#
# License https://www.backblaze.com/using_b2_code.html
#
######################################################################

import logging
import re
```

```
logger = logging.getLogger(__name__)

class RegexSet(object):
    """
    Holds a (possibly empty) set of regular expressions, and knows how to
     check
    whether a string matches any of them.
    """

    def __init__(self, regex_iterable):
        self._compiled_list = [re.compile(r) for r in regex_iterable]

    def matches(self, s):
        return any(c.match(s) is not None for c in self._compiled_list)

class ScanPoliciesManager(object):
    """
    Policy object used when scanning folders for syncing, used to decide
    which files to include in the list of files to be synced.
    """

    def __init__(
        self,
        exclude_dir_regexes=tuple(),
        exclude_file_regexes=tuple(),
        include_file_regexes=tuple(),
    ):
        # [TODO]

    def should_exclude_file(self, file_path):
        """
        Given the full path of a file, should it be excluded from the
        scan?

        :param file_path: The path of the file, relative to the root
        directory
                          being scanned.
        :return: True iff excluded.
        """
        return self._exclude_file_set.matches(file_path) and \
                not self._include_file_set.matches(file_path)

    def should_exclude_directory(self, dir_path):
        """
        Given the full path of a directory, should all of the files in it
         be
        excluded from the scan?

        :param dir_path: The path of the directory, relative to the root
        directory
                          being scanned.  The path will never end in '/'.
        :return: True iff excluded.
        """
        return self._exclude_dir_set.matches(dir_path)

DEFAULT_SCAN_MANAGER = ScanPoliciesManager()

```
--- END FILE ---
```

```
Please first localize the positions to edit based on the task statement,
and then output the files that need to be deleted, modified or added.
'- file' means deleting the file;
'* file' means modificating the file;
'+ file' means adding the file.
The file should list like below:
```
- file1.py
* file2.py
* file3.py
+ file4.py
```

To solve the task, you should then generate *SEARCH/REPLACE* edits.

Every *SEARCH/REPLACE* edit must use this format:
1. The file path
2. The start of search block: <<<<<<< SEARCH
3. A contiguous chunk of lines to search for in the existing source code
4. The dividing line: =======
5. The lines to replace into the source code
6. The end of the replace block: >>>>>>> REPLACE

Here is an example:

```python
### mathweb/flask/app.py
<<<<<<< SEARCH
from flask import Flask
=======
import math
from flask import Flask
>>>>>>> REPLACE
```

Please note that the *SEARCH/REPLACE* edit REQUIRES PROPER INDENTATION.
If you would like to add the line '          print(x)', you must fully
write that out, with all those spaces before the code!
Wrap the *SEARCH/REPLACE* edit in blocks ```python...```.
When multiple edits should be done, please output *SEARCH/REPLACE* edit
one by one and give your reasoning process before each *SEARCH/REPLACE*
block.

>>>>>>>>>>>>>>>>>>> assistant:
The plan of solving this software task:
The task requires completing the '__init__' method of the '
ScanPoliciesManager' class in the 'b2/sync/scan_policies.py' file. The
method should initialize the class attributes using the provided regular
expression tuples. The class uses 'RegexSet' objects to manage and check
regular expressions. Therefore, the '__init__' method should create '
RegexSet' instances for each of the regular expression tuples: '
exclude_dir_regexes', 'exclude_file_regexes', and 'include_file_regexes'.

The files that should be edited include:
* b2/sync/scan_policies.py

Here are my edits for code.

The motivation of this edit:
The 'ScanPoliciesManager' class is designed to manage policies for
scanning directories and files, determining which should be included or
excluded based on regular expressions. The '__init__' method needs to
initialize the class attributes '_exclude_dir_set', '_exclude_file_set',
```

```
and `_include_file_set` using the `RegexSet` class. This setup allows the
 `should_exclude_file` and `should_exclude_directory` methods to function
 correctly by checking file and directory paths against the compiled
regular expressions.

```python
### b2/sync/scan_policies.py
<<<<<<< SEARCH
        exclude_file_regexes=tuple(),
        include_file_regexes=tuple(),
    ):
        # [TODO]

    def should_exclude_file(self, file_path):
        """
=======
        exclude_file_regexes=tuple(),
        include_file_regexes=tuple(),
    ):
        self._exclude_dir_set = RegexSet(exclude_dir_regexes)
        self._exclude_file_set = RegexSet(exclude_file_regexes)
        self._include_file_set = RegexSet(include_file_regexes)

    def should_exclude_file(self, file_path):
        """
>>>>>>> REPLACE
```
```

