# OpenReview forum: "SWE-Ext: Extending and Scaling Augmented Data for Repository-Level Coding Tasks"
_ICLR.cc/2026/Conference — ICLR 2026 Conference Withdrawn Submission_

### Official Review · Reviewer_gZGa · 2025-10-21

**Soundness:** 3
**Presentation:** 3
**Contribution:** 3
**Rating:** 6
**Confidence:** 3

**Summary:**

SWE-Ext proposes a scalable pipeline to extend augmented training data for repository-level coding beyond the current Python-only issue-fixing focus by adding multilingual pull-request data (10 languages) and an auxiliary code-completion task. The dataset factorizes supervision along four stages of an agentless pipeline: file localization, component localization, code editing, and code completion. GPT-4o is leveraged as the expert model to synthesize training data. Hints from groundtruth are provided when the expert model fails. Experiments show consistent downstream gains on Python-only benchmarks (SWE-Bench, FEA-Bench). In particular, training on multilingual and completion data both enhances performance on Python tasks. Scaling analysis shows predictable gains across model sizes (7B→32B) and data volume when applying these extensions.

**Strengths:**

* SWE-Ext presents a scalable way of synthesizing training data for software engineering tasks, by leveraging supervision from groundtruth to steer the expert model to generate correct samples. The method does not rely on runtime environments, significantly reducing the cost.
* Experiments demonstrate clear gains on SWE-Bench Verified and FEA-Bench. Ablations show that training on multilingual or code completion data can both improve the performance on python-only SWE tasks, proving that those are both promising approaches for data scaling.
* The paper is written clearly and easy to follow.

**Weaknesses:**

* Existing data augmentation frameworks with executable environments (e.g., swe-gym, swe-smith) are generally scaffold-agnostic, in the sense that one can recreate training trajectories for any agent scaffold by running rejection sampling. However, I do not see a straightforward way to redo the exercise of SWE-Ext for a different agent, for example, OpenHands, as it may require a separate design to inject hints.

* The use of hints in the form of groundtruth leakage may hurt data quality. In some cases where LLMs do not understand the full path towards the correct final answer, they may hallucinate the intermediate reasoning steps. Did you observe such cases, and how were they handled?

**Questions:**

* What motivates you to choose an agentless scaffold for this work instead of an end-to-end multi-turn agent with tools?
* Is the $P$ in L077 meant to be $L$?
* L144 seems missing an "and".

---

> ### Author Response · Authors · 2025-12-02
> **Response to Reviewer gZGa**
>
> We thank the reviewer for the positive evaluation and thoughtful comments.
>
> ---
>
> ### **1. Limited scaffold-agnosticity vs. executable-data frameworks**
>
> Execution-based frameworks (e.g., SWE-Gym, SWE-Smith) naturally allow scaffold-agnostic rejection sampling. In contrast, SWE-Ext focuses on **static augmented data**, which is intentionally designed as a *mid-training* stage. While reusing SWE-Ext for different agents may require lightweight adapter rules, we view this as complementary to, but not a replacement for execution-based pipelines.
>
> ---
>
> ### **2. Potential hallucination from hint-based augmentation**
>
> We acknowledge that hint injection may occasionally lead to inconsistent reasoning chains. Our rule-based filtering (e.g., patch–context consistency checks, structural validation) removes most malformed samples. However, this is an efficient way than using rejective sampling on iterative agents like OpenHands.
>
> ---
>
> ## **Responses to Questions**
>
> **Why use an agentless scaffold instead of a multi-turn tool-using agent?**
> Because our goal is to study **scalable static augmentation**, which does not rely on execution tools. Agentless scaffolds allow us to analyze data effects without confounding from complex interactive behaviors. Dynamic tool-using agents can be added later via rejection-sampling SFT or RL. Also, the agentless scaffold is at a low cost and can be run under the 32k context length, while multi-turn agent, like openhands, will reach the context limit easily after just a few rounds.
>
> **Typo in L077?**
> Yes, this will be corrected.
>
> **Missing “and” in L144?**
> We appreciate the catch and will fix it.

---

### Official Review · Reviewer_Gpoy · 2025-10-26

**Soundness:** 2
**Presentation:** 3
**Contribution:** 2
**Rating:** 6
**Confidence:** 2

**Summary:**

This paper proposes SWE-Ext, a framework for extending repository-level datasets by adding multilingual and code-completion data. It decomposes software engineering tasks from GitHub pull requests into subtasks (localization, editing, completion) and uses GPT-4o as a data generation model to fine-tune Qwen2.5-Coder models. Experiments show consistent but moderate improvements on SWE-Bench and FEA-Bench benchmarks.

**Strengths:**

The paper proposes a well-structured and scalable data collection process for repository-level code tasks. It leverages real GitHub PRs, ensuring realism and high coverage.

Introducing multilingual and completion-based augmentation is original in the context of SWE-Bench-like setups, addressing the monolingual bias of prior datasets (mostly Python).

SWE-Ext offers a potentially reusable dataset and pipeline for training and benchmarking LLMs in real-world software engineering contexts, without requiring execution-based data collection.

**Weaknesses:**

The dataset relies entirely on augmented supervision from PRs and expert model annotations, without verification through execution or testing (unlike SWE-Gym or R2E). This limits the data’s reliability and may include noisy or incomplete samples.

The reported improvements (+1–2% on SWE-Bench, +2.5% on FEA-Bench) are modest relative to the scale of data expansion. The paper does not fully analyze the efficiency trade-off versus dataset size.

Despite the multilingual data generation, benchmarks are only in Python, so cross-lingual generalization is inferred rather than directly measured.

The manuscript could benefit from clearer explanation in the experiments section, especially the description of the CosAgentless system and training configurations.

**Questions:**

Have you tested zero-shot performance on non-Python repository-level tasks to confirm actual multilingual transfer rather than auxiliary effects?

How do you filter out low-quality PRs or synthetic reasoning errors from the GPT-4o expert outputs? Any human validation?

Could SWE-Ext be combined with verification methods like SWE-Gym or SWE-Smith to yield hybrid supervision?

---

> ### Author Response · Authors · 2025-12-02
> **Response to Reviewer Gpoy**
>
> We thank the reviewer for the positive assessment and helpful suggestions.
>
> ---
>
> ### **1. Static data without execution verification**
>
> Our goal is to explore **scalable augmented static data** as a complementary path to execution-based pipelines. While static data may contain some noise, it offers **much lower cost and broader coverage**, and can serve as an effective **mid-training stage** before applying rejection-sampling SFT or RL.
>
> ---
>
> ### **2. Modest empirical gains**
>
> The underlying GitHub PR corpus is already large, placing the model on the **diminishing-returns region** of the scaling curve. Even small improvements indicate that the extensions help utilize additional data more effectively.
>
> ---
>
> ### **3. Multilingual data vs. Python-only benchmarks**
>
> Our intent is to leverage multilingual data to increase **diversity** under fixed PR sources, not to perform cross-lingual evaluation. For  the methods targeting at multilingual evaluation, we will address in the future work.
>
> ---
>
> ### **4. Experimental clarity**
>
> We appreciate the suggestion and will refine the description of the CosAgentless system and training setup for improved clarity. It is a system that replace the localization process with an iterative watch-select process.
>
> ---
>
> ## **Questions**
>
> **Zero-shot non-Python SWE tasks?**
> Not yet. Large-scale cross-language SWE benchmarks are not available, but we plan to explore this direction.
>
> **Filtering noisy PRs / expert annotations?**
> We apply rule-based filtering (e.g., structural validity checks, patch consistency) but no human annotation due to scale. We will add details in the revision.
>
> **Combining SWE-Ext with verification-based methods?**
> SWE-Ext is designed to be compatible with **rejection-sampling SFT or RL**, and hybrid supervision is a natural next step. We will highlight this as future work.

---

### Official Review · Reviewer_nvks · 2025-10-29

**Soundness:** 2
**Presentation:** 2
**Contribution:** 2
**Rating:** 2
**Confidence:** 3

**Summary:**

This paper proposes two data augmentation methods, multilingualism and additional code completion tasks, for SWE-Bench-like tasks. Based on these two augmentation methods, this paper proposes a new training dataset, SWE-Ext. Trained on SWE-Ext, models perfoms well on SWE-Bench and FEA-Bench.

**Strengths:**

1. The studied topic is valuable.
2. The paper is well-written and easy to follow.
3. Training with the proposed SWE-Ext, the performance increases.

**Weaknesses:**

1. The experiments are not comprehensive. There are three experimental settings in the paper: 1) without augmentation 2) with multilingual 3) with code completion. Though both 2) and 3) are shown to have a positive influence on performance, there are no experimental results of the combination of 2) and 3).
2. The two proposed augmentation methods are quite common in coding-related tasks, which limits the novelty of the paper.
3. The authors only train Qwen series LLMs with SWE-Ext. More results from other LLMs are needed to demonstrate the general benefit of SWE-Ext.
4. For FEA-Bench, the authors only compare their models with general LLMs. More results from specific coding LLMs are needed for a more comprehensive evaluation.
5. I don't agree that the experimental settings and results support the claim that multilingualism augments LLM training. The further improvement of performance may not come from the multilingualism of the data, but rather from the diversity of the data. A more rigorous setting is to translate exactly the same python projects to other programming languages.

**Questions:**

Please see my comments above.

---

> ### Author Response · Authors · 2025-12-02
> **Response to Reviewer nvks**
>
> We thank the reviewer for their helpful comments. Below we address each concern in order.
>
> ---
>
> ## **1. Missing combined results for multilingual + code completion**
>
> We acknowledge the reviewer’s concern. Due to limited computational resources, we were unable to train the **32B** model with both extensions combined. However, we did include combined experiments on the **7B** model: the **scaling curves in Figure 3(a)** are obtained using training data that incorporates **both** multilingual and completion-based extensions.
>
> ---
>
> ## **2. The proposed augmentation methods seem common**
>
> While multilingual extension and completion-style augmentation have appeared in *script-level code generation*, there is **no prior work in the SWE domain** that systematically studies how **extended data** affects performance under *single-language evaluation*.
> Moreover, the reviewer may have misunderstood our intent: our contributions are **not proposing two augmentation methods**, but proposing **two extension strategies for augmented data construction**. The augmented data is a easy and low-cost path for scaling, compared to verified data.
>
> ---
>
> ## **3. Only training Qwen-series LLMs**
>
> Training SWE models requires **extremely long context windows** and is computationally expensive. Existing works, such as **SWE-Smith** also train exclusively on **Qwen2.5-Coder models** for this reason. Therefore, using Qwen models does not limit the generality of our findings and is consistent with community practice. We will clarify this in the revision.
>
> ---
>
> ## **4. FEA-Bench comparisons include only general LLMs**
>
> For FEA-Bench, we report results following the availability of **publicly reported numbers** in prior work. Many specialized coding LLMs do not provide FEA-Bench results, which limits the comparability.
> In addition to our own models, we consistently report published numbers to maintain fairness. We will make this clearer in the revision.
>
> ---
>
> ## **5. Concern about whether improvements come from multilingualism or general data diversity**
>
> Our goal is precisely to investigate how to **maximize data utilization** when the amount of GitHub PR source data is fixed. Multilingual extension naturally increases **data diversity**, which we believe is just the *mechanism* through which it benefits model performance. This point aligns with our intended research question.
>
> Translating the exact same large-scale Python repositories to other languages, is unfortunately **not feasible** in practice due to the size and complexity of real-world codebases. We will clarify this limitation in the paper.

---

### Official Review · Reviewer_QqW5 · 2025-11-01

**Soundness:** 1
**Presentation:** 2
**Contribution:** 1
**Rating:** 2
**Confidence:** 4

**Summary:**

This paper introduces SWE-Ext, a pipeline for constructing training data for repository-level coding tasks. The authors extend existing approaches along two key dimensions: multilingual coverage (spanning 10 programming languages) and an auxiliary code completion task. The pipeline processes GitHub pull requests to create four complementary datasets targeting file localization, component localization, code editing, and code completion. Through experiments on SWE-Bench and FEA-Bench using Qwen2.5-Coder models (7B and 32B), the authors demonstrate that multilingual data improves localization and overall performance, while code completion data primarily enhances editing capabilities.

**Strengths:**

+ Collecting training data to solve repository-level SWE tasks is an important topic.

**Weaknesses:**

**Unsatisfactory Performance compared to closely related work**

The best trained SWE-Ext-32B models report 32.6% on SWE-Bench Verified, which is notably worse than the most directly comparable baselines: SWE-Smith-32B (40.2%) and R2E-Gym-32B (34.4%). These two methods are both training data construction approaches using the same 32B model scale, making them the most relevant comparisons for evaluating SWE-Ext's data augmentation strategy. The performance gap of 7.6% and 1.8% respectively suggests that SWE-ext may be less effective than existing data construction techniques, such as SWE-smith and R2E-Gym.

**Missing some 32B baselines**

The paper misses recent strong 32B baselines that have demonstrated significantly better performance on SWE-Bench Verified. Notably, DeepSWE [1] and SWE-Swiss [2] report substantially higher performance on the same benchmark. Without comparing against these strong baselines, it is difficult to assess where SWE-Ext stands relative to the strong baselines in swe-bench at similar model scales.

[1] DeepSWE: https://www.together.ai/blog/deepswe

[2] SWE-Swiss: https://github.com/zhenyuhe00/SWE-Swiss/tree/main

**Lack of controlled apple-to-apple comparison.**

Importantly, as a training data construction work, the paper fails to provide an apple-to-apple comparison that isolates the effectiveness of the data itself. The comparisons with SWE-Smith, SWE-Gym, and R2E-Gym involve different base models (e.g., SWE-Smith uses different expert models for data generation), different agent scaffolds (OpenHands vs. Agentless vs. CosAgentless), and different system configurations. To truly validate the contribution of SWE-Ext's multilingual and completion-based data augmentation, the authors should conduct controlled experiments where only the training data varies while keeping the base model, agent system, and inference setup constant. For instance, training on SWE-Smith's data versus SWE-Ext's data using the same Qwen2.5-Coder-32B model and CosAgentless system would provide a direct assessment of data quality. Without such controlled comparisons, it remains unclear whether the observed performance differences stem from superior data construction, better agent design, or simply different experimental configurations.

**Marginal Improvements from Multilingual and Code Completion Extensions**

The highlighted contributions of extending data to multiple languages and incorporating code completion tasks yield only marginal improvements that raise questions about their practical value. The multilingual extension provides only +1.4% improvement (31.2% → 32.6%) on SWE-Bench. Similarly, the code completion extension shows limited impact with only +1.0% improvement (31.2% → 32.2%) on SWE-Bench Verified, and notably comes at the cost of degraded localization performance (component Hit@1 drops from 55.0% to 52.8% in Table 4), suggesting misalignment with the full repository-level coding pipeline. The paper lacks critical analysis on whether combining both extensions yields additive improvements, whether there are diminishing returns, or whether simply collecting more high-quality Python issue-resolution data would be more efficient.

**Static Data Collection Misaligned with Dynamic Agent Paradigms and Lacks Practical Extensibility**

The paper's purely static data collection approach is fundamentally misaligned with the dynamic, interactive nature of modern LLM-based coding agents, limiting both its effectiveness and practical applicability. Current state-of-the-art coding agents rely heavily on iterative interactions with execution environments, incorporating feedback from test runs, linting errors, compilation failures, and runtime observations to guide code modifications. In contrast, SWE-Ext constructs training data entirely from static GitHub pull requests without any execution-based verification or tool-using, essentially restrict the training recipe from dynamic agentic capabilities.

Moreover, it seems practically infeasible to extend SWE-ext with dynamic agentic trajectories, since it does not provide unified configuration interface for too much varied repositores that span across multiple languages. This restriction makes the value of SWE-ext even more downgraded.

**Questions:**

- Could you provide an apple-to-apple comparison where you train models on SWE-Smith's data, R2E-Gym's data, and SWE-Ext's data using the exact same base model, agent scaffold, and inference configuration? This would isolate the contribution of your data construction approach from confounding factors like different agent scaffolds or model choices.

- Could the authors add comparison with DeepSWE and SWE-Swiss?

- Could the author explain why they focus on static trajectories without any dynamic interactions and tool using?

---

> ### Author Response · Authors · 2025-12-02
> **Response to Reviewer QqW5**
>
> We thank the reviewer for the constructive feedback. Our work aims to investigate **augmented static data as an alternative data construction path for SWE**, which we believe is complementary to the dominant line of works that rely on **verifiable, execution-based data** (L47). Under the difficulty of scaling execution environments to large codebases and multi-language repositories, our goal is to show that *static augmented data remains a valuable and scalable direction*, and can serve as a strong foundation for downstream **rejection-sampling SFT** or **RL fine-tuning**.
>
> Below we address each concern in order.
>
> ---
>
> ## **1. Unsatisfactory performance compared to closely related work**
>
> As shown in Table 2 and discussed around L318–L321, our model achieves **competitive performance under limited resources**. The reviewer’s cited baselines, SWE-Smith-32B (40.2%) and R2E-Gym-32B (34.4%), are marked in Table 2 as using **execution environments (Exec)**, which substantially increases data quality by filtering with verifiable outcomes.
> We also emphasize (L321) that **models with higher performance either (i) use significantly larger model scales, or (ii) are distilled from expert LLMs that themselves exhibit much higher success rates**. In contrast, SWE-Ext focuses on *scalable data augmentation without execution*, making its performance competitive given the constraints.
>
> ---
>
> ## **2. Missing some 32B baselines**
>
> We appreciate the reviewer pointing this out.
>
> * **DeepSWE** is a method built on *verifiable trajectories* and therefore belongs to a different training paradigm from SWE-Ext; it is not the primary comparison target for a study on *static augmented data*.
> * **SWE-Swiss** currently exists only as a blog post without a citable academic reference. For this reason, it was not included, but we will mention it in the revised version for completeness.
>
> ---
>
> ## **3. Lack of controlled apple-to-apple comparison**
>
> We agree that controlled comparisons are ideal. Our comparison table follows the setup used in **Table 3 in SWE-Smith paper**, which similarly compares across different data-construction pipelines, base models, and agent scaffolds due to the **high resource cost** of building fully controlled SWE training pipelines. For SWE task, the most realistic apple-to-apple evaluation is through **ablations**, which we already include in Section 5. These ablations isolate the effects of multilingual extensions, completion-based extensions, and PR-derived data.
>
> Moreover, by comparing SWE-Ext with other **static-data approaches** such as SWE-Fixer, SoRFT, and MCTS-Refine (Table 2), we demonstrate the **effectiveness of our augmentation strategy** independent of agent or environment differences.
>
> ---
>
> ## **4. Marginal improvements from multilingual and completion extensions**
>
> The relative gains appear small mainly because the underlying GitHub PR corpus is already **large**, placing our model on the **diminishing-returns region of the scaling curve**. Section 5.3 (L400–L404) explicitly shows that **under fixed source PR data**, our extensions improve the scaling behavior of the model.
>
> Even if the incremental improvements are moderate, the extensions are **important for fully utilizing available data**: discarding multilingual or completion-style data would unnecessarily reduce the effective dataset size. In high-data regimes, extracting any additional signal—however small—is still valuable.
>
> ---
>
> ## **5. Static data collection misaligned with dynamic agent paradigms**
>
> Our work deliberately explores **static augmented data**, not verifiable trajectories, for two reasons:
>
> 1. **Scalability and cost-efficiency**: Rule-based static augmentation requires **no execution environment**, avoiding the high cost of rejection sampling or dynamic tool-use. This makes the method applicable to diverse repositories across multiple languages.
>
> 2. **Complementarity, not replacement**: SWE-Ext is intended to act as a form of **mid-training**, providing broad coverage and scalable pretraining signals. Dynamic paradigms (e.g., rejection-sampling SFT or RL) can be applied **after** SWE-Ext to further align the model with interactive tool-using behaviors.
>
> We agree that dynamic trajectories are valuable, but SWE-Ext addresses a different point in the training pipeline, offering a **scalable static foundation** on top of which dynamic methods can be further applied.

---

### Note · Authors · 2026-01-06

I have read and agree with the venue's withdrawal policy on behalf of myself and my co-authors.